# Understanding and Mitigating the Label Noise in Pre-training on Downstream Tasks

**Hao Chen**[1,2]*, **Jindong Wang**[2]†, **Ankit Shah**[1], **Ran Tao**[1],
**Hongxin Wei**[3], **Xing Xie**[2], **Masashi Sugiyama**[4,5], **Bhiksha Raj**[1,6]

[1]Carnegie Mellon University, [2]Microsoft Research Asia, [3]SusTech,
[4]RIKEN AIP, [5]The University of Tokyo, [6]Mohamed bin Zayed University of AI

## Abstract

Pre-training on large-scale datasets and then fine-tuning on downstream tasks have become a standard practice in deep learning. However, pre-training datasets, while inaccessible or too expensive to handle, often contain label noise that may adversely affect the generalization of the model and pose unexpected risks. This paper aims to understand the nature of noise in pre-training datasets and then mitigate its impact on downstream tasks. Specifically, through extensive experiments of supervised pre-training models on synthetic noisy ImageNet-1K and YFCC15M datasets, we demonstrate that while slight noise in pre-training can benefit in-domain (ID) performance, where the training and testing data share the same distribution, it always deteriorates out-of-domain (OOD) performance, where training and testing distributions are different. We empirically ascertain that the reason behind is noise in pre-training shapes the feature space differently. We then propose a light-weight black-box tuning method (NMTune) to affine the feature space to mitigate the malignant effect of noise and improve generalization on both ID and OOD tasks, considering that one may not be able to access or fully fine-tune the pre-trained models. We conduct extensive experiments on popular vision and language models including APIs that are supervised and self-supervised pre-trained on real data for evaluation. Our results show the importance of this novel and fundamental research direction, which we term *Noisy Model Learning*.

## 1 Introduction

The transfer learning paradigm of pre-training and fine-tuning (PT-FT) (Kornblith et al., 2019) has become the de facto standard in today's deep learning research and application. Instead of training a neural network from scratch for each individual task, which can be time-consuming, resource-intensive, and less adaptable, the PT-FT paradigm first pre-trains a relatively larger and more general model with huge volumes of datasets, and then transfers this pre-trained model (or the foundation model (Bommasani et al., 2021)) to various downstream tasks (He et al., 2019; Radford et al., 2021; He et al., 2022; Brown et al., 2020). For instance, ResNet (He et al., 2016a) and Vision Transformers (Dosovitskiy et al., 2020) pre-trained on ImageNet (Russakovsky et al., 2015) and larger but potentially noisy datasets (Kolesnikov et al., 2020; Xie et al., 2020; Ridnik et al., 2021) have been widely adopted in computer vision. The PT-FT paradigm has also become predominant in natural language processing (Devlin et al., 2018; Liu et al., 2019; Radford et al., 2018; 2019; Brown et al., 2020; OpenAI, 2023; Touvron et al., 2023a) and multi-modality (Radford et al., 2021; Schuhmann et al., 2022), where the pre-training is usually on large datasets scraped from the web.

The generalization and transferability of the pre-trained models are usually not guaranteed to be satisfying on downstream tasks, and the reason can lie in either the pre-training or the fine-tuning. Over the years, there have been tremendous efforts in improving the performance of fine-tuning in various practical downstream scenarios: out-of-distribution generalization (Chen et al., 2021; Kumar et al., 2022), semi-supervised learning (Sohn et al., 2020; Wang et al., 2022), imbalanced learning (Zhang et al., 2023b; Wang et al., 2023b), noisy label learning (Song et al., 2022; Li et al., 2022),

---

*haoc3@andrew.cmu.edu, work done during a research intern at MSRA.
†Correspondence to: jindong.wang@microsoft.com

to name a few. While it is a common belief that scaling up the size of the pre-training data can benefit the downstream performance (Kaplan et al., 2020), its distribution also plays an essential role (Entezari et al., 2023; Zhang et al., 2023a). Recently, Nguyen et al. (2022) and Lee et al. (2022) found that the *quality* of the pre-training data is more important for robust generalization compared to the quantity. The bias in pre-training data created by the collection (and annotation) process, e.g., corrupted, poisoned, and false information (Blodgett & O'Connor, 2017; Chang et al., 2020), can also impose malicious and unexpected influence to downstream tasks (Bommasani et al., 2021).

Take label noise as an example. Training CLIP (Radford et al., 2021) on LAION-2B (Schuhmann et al., 2022), which is a billion-scale uncurated image-text pair dataset, can just match the performance of training it on WIT-400M (Radford et al., 2021), which is heavily cleaned and processed by OpenAI. The label noise in large-scale datasets inevitably exists owing to the data collection process by human annotators and web crawlers. It thus can be difficult to avoid or eliminate in pre-training (Ridnik et al., 2021; Vasudevan et al., 2022; Schuhmann et al., 2022). In fact, there are already numerous models pre-trained on large-scale noisy data and have been transferred on downstream tasks, such as Noisy Student (Xie et al., 2020), BiT (Kolesnikov et al., 2020), and Open CLIP (Cherti et al., 2023). Not to mention the enormous but noisy raw text (Yang et al., 2019; Lee et al., 2022) that has been utilized to pre-train language models such as BERT (Devlin et al., 2018) and GPT (Radford et al., 2019; Brown et al., 2020). As the pre-trained models and datasets have been growing significantly, it has become increasingly important and challenging to understand *how the noise in pre-training data affects the performance of pre-trained models on downstream tasks.*

This paper presents the first study on this unexplored problem, demystifying the label noise in pre-training data, understanding its effects on downstream tasks, and then mitigating such (malignant) effects. Notably, there are existing efforts under the name of "noisy label learning" that train a robust model *given* noisy training data (Ghosh et al., 2017; Li et al., 2020; Northcutt et al., 2021). Our problem is inherently different since the noisy labels exist in the (usually black-box) pre-training data, and we do not make noise assumptions on the downstream data (while they can be used together as in Section 4.3; more discussion is in Section 5). Due to the increasing size of pre-trained models and datasets, it becomes notoriously difficult to alter the pre-training process or fine-tune the entire models (black-box or cannot be updated due to the large parameter size and the constrained computation).[1] Therefore, given a pre-trained model, we should take special care of the *fine-tuning* to overcome the influence of noise in pre-training on downstream tasks.

Our study aims to answer the following key questions: 1) *Influence:* Does the noise in pre-training data have an influence on downstream performance? 2) *Analysis:* Why does such influence happen? and 3) *Mitigation:* How to mitigate such influence in a light-weight and black-box fine-tuning process? We present an in-depth analysis to answer the above questions, based on the popular *supervised* pre-training paradigm.[2]

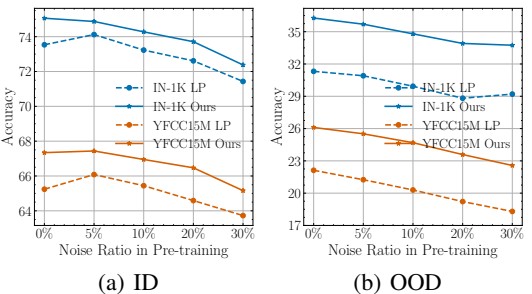

(a) ID  (b) OOD

Figure 1: In-domain (ID) and out-of-domain (OOD) downstream performance when supervised pre-training the model on synthetic noisy ImageNet-1K (IN-1K) and YFCC15M of various noise ratios. We compare linear probing (LP) and the proposed method on 14 ID and 4 OOD tasks. On ID, 5% noise in pre-training benefits the LP performance. Our method not only boosts the general performance but also rectifies the model pre-trained on clean data to be comparable to 5% noise. On OOD, noise in pre-training is detrimental to robustness performance when conducting LP. Our method improves the transferability on OOD tasks significantly compared to LP.

- **Influence: The label noise in pre-training data has both benevolent and malignant influence on downstream tasks.** In Sections 2.1 and 2.2, we conduct realistic experiments

---

[1]Llama (Touvron et al., 2023a;b) model requires multiple V100 GPUs to fine-tune, which is not affordable to most ordinary researchers; and proprietary models like ChatGPT cannot be locally fine-tuned.

[2]Supervised and self-supervised learning are the most popular pre-training schemes. The former learns the mapping from inputs to labels (He et al., 2016a; Radford et al., 2021), while the latter does not rely on labels, but predicts parts of the data itself (Devlin et al., 2018; He et al., 2022).

with ResNet-50 models (He et al., 2016a) fully-supervised and contrastive pre-trainied on synthetic noisy ImageNet-1K and YFCC15M (Thomee et al., 2016) with various noisy ratios $(0, 5\%, 10\%, 20\%, 30\%)$ and then study the generalization performance on the downstream in-domain (ID) and out-of-domain (OOD) tasks. We observe that, on ID tasks, slight noise (up to $5\%$ or $10\%$) can benefit generalization performance. In contrast, even $5\%$ noise can drastically deteriorate robustness and transferability on OOD tasks, as shown in Figure 1 and Figure 2.

- **Analysis: The label noise in pre-training shapes the feature space significantly of the pre-trained model.** In Section 2.3, we conduct empirical analysis from the singular value spectrum on the feature space of the pre-trained models. Noise in pre-training results in the decreasing largest singular value and flatter singular value distribution with a higher dimension span in the feature space. An initial increase in the spanning dimension of the feature space is beneficial to the discriminability on ID tasks. Still, it then becomes detrimental with the further increase, indicating more feature capacities are learned to fit to noise structure. The decrease in the dominant singular values leads to less transferability for OOD tasks (Chen et al., 2019), as shown in Figure 3.

- **Mitigation: We design a simple black-box fine-tuning algorithm to reshape the pre-trained feature space, reducing the influence of noisy pre-training data and boost the performance of downstream tasks.** In Section 3, based on the analysis, we propose three regularization objectives on the singular value spectrum that help affine the feature space. We demonstrate the effectiveness of the proposed method on noisy ResNet-50 models with extensive analysis, as shown in Figure 1. In Section 4, we further validate our method on popular noisy pre-trained models (and APIs) and present superior generalization performance for both vision and language tasks.

Beyond our analysis, we view this research as a novel and complementary topic to the classic noisy label learning setting, termed as *Noisy Model Learning* (NML). We think the value of this direction is even more significant in the era of large foundation models (Bommasani et al., 2021), where the downstream users only have access to the model weights or APIs. It would be of particular interest to explore how to eliminate the malignant influence of noise in pre-training on downstream tasks when adapting these models without full fine-tuning, since it may exist in broader applications such as the detection and segmentation in medical and autonomous driving. We hope that future research on this topic can facilitate a better understanding and application of large foundation models.

## 2 Understanding the Label Noise in Pre-trained Models

In this section, we empirically and systemically investigate the effect of noisy labels in the supervised pre-training on the learned representations. We build our evaluation and analysis on the realistic motivating experiments of training ResNet-50 (He et al., 2016a) on synthetic noisy ImageNet-1K (Russakovsky et al., 2015) and YFCC15M (a subset of YFCC100M (Thomee et al., 2016)).

### 2.1 Experiments Design

**Noisy pre-training datasets**. We assume that the supervised pre-training dataset consists of inputs $\mathbf{x} \sim \mathcal{X}$ and supervisions $y \sim \mathcal{Y}$. We define a clean dataset $\mathcal{D} = \{(\mathbf{x}_i, y_i)\}_{i \in [N]}$ of size $N$ with accurate supervisions, where $[N] := \{1, \ldots, N\}$. We assume that $y$ can exist in different formats in pre-training, e.g., an actual label for the input as in fully-supervised learning (Russakovsky et al., 2015; He et al., 2016a; Ridnik et al., 2021) or a text description for an input image as in contrastive learning of CLIP (Thomee et al., 2016; Radford et al., 2021; Jia et al., 2021; Changpinyo et al., 2021; Desai et al., 2021; Schuhmann et al., 2021; 2022). Due to the scale of data collection and the cost of data annotation, the pre-training dataset can usually contain noisy supervision $\hat{y}$ that does not accurately match the corresponding $\mathbf{x}$ (Recht et al., 2019; Beyer et al., 2020; Northcutt et al., 2021; Yun et al., 2021; Vasudevan et al., 2022; Schuhmann et al., 2022). We define such noisy pre-training dataset as $\hat{\mathcal{D}} = \{(\mathbf{x}_i, \hat{y}_i)\}_{i \in [N]}$ and the noise ratio $\gamma$ as the percentage of noisy supervision in $\hat{\mathcal{D}}$.

**Pre-trained models**. The pre-trained models serve as a foundation for downstream tasks and usually can be abstracted as the stack of a feature extractor and a projection head. We define the feature extractor with learned parameters $\phi$ as a mapping function $f_\phi$ from the input space to feature space of dimension $D$: $f_\phi : \mathcal{X} \rightarrow \mathcal{F}$. The projection head $g_\theta : \mathcal{F} \rightarrow \mathcal{Y}$ is jointly pre-trained with the feature extractor, but not used when adapting $f_\phi$ on downstream tasks. We consider two types of supervised pre-training on images for this motivating example: fully supervised pre-training where $y$ is the actual class label and the projection head is a linear classifier (He et al., 2016a), and contrastive pre-

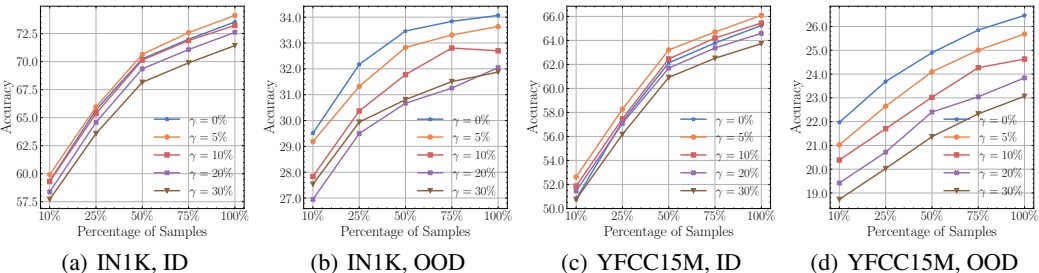

(a) IN1K, ID     (b) IN1K, OOD     (c) YFCC15M, ID     (d) YFCC15M, OOD

Figure 2: Average ID and OOD evaluation results of ImageNet-1K (IN-1K) fully supervised pre-training ((a) and (b)) and YFCC15M CLIP pre-training ((c) and (d)) on downstream tasks with various percentages of data using ResNet-50. On ID evaluation, the transfer performance first increases as noise increases (to $5\%$ or $10\%$) and then decreases with more noise. On OOD evaluation, the robustness performance constantly decreases once noise is introduced in pre-training.

training with text supervision (CLIP) where $y$ is the text and the projection is a non-linear function maps the image and text to a common feature space (Radford et al., 2021; Cherti et al., 2023).

**In-domain (ID) and out-of-domain (OOD) evaluation**. To investigate the effect of noisy supervision comprehensively, we leverage both in-domain (ID) and out-of-domain (OOD) evaluation to assess the generalization capability of the pre-trained feature extractor (Djolonga et al., 2021) $f_\phi^\gamma$ that are obtained from the pre-training data of different noise ratios. To evaluate the pre-trained models on a downstream dataset $\mathcal{D}' = \{(x_i, y_i)\}_{i \in [M]}$[3] and measure the quality of the learned representation, we conduct linear probing (LP)[4], where only a $C$-way linear classification head is re-trained on the downstream dataset and the feature extractor is frozen. The linear probing can be viewed as a simple black-box tuning method for pre-trained models that are typically large and difficult or unable to fully fine-tune. For ID evaluation, we assume the same marginal distribution over $\mathcal{X}$ for both training and testing. In contrast, for OOD evaluation, we train the linear classifier on a source distribution and evaluate it on (multiple) different target distributions (Kumar et al., 2022).

**Experiment setup**. We use ImageNet-1K (IN-1K) (Russakovsky et al., 2015) in fully supervised pre-training and YFCC15M (Thomee et al., 2016) in CLIP pre-training, with ResNet-50 (He et al., 2016a). To introduce noisy supervision in the datasets, we uniformly flip the ground truth class label into the other classes in IN-1K and randomly swap the text description from another image-text pair in YFCC15M. We set the noise ratio $\gamma$ to $\{0\%, 5\%, 10\%, 20\%, 30\%\}$, where $0\%$ represents the clean dataset. For ID evaluation, we use 14 downstream datasets including CIFAR-10/100 (Krizhevsky et al., 2009), Flowers102 (Nilsback & Zisserman, 2008), Food101 (Bossard et al., 2014), Oxford-Pet (Parkhi et al., 2012), StanfordCars (Krause et al., 2013), FGVCAircraft (Maji et al., 2013), SVHN (Netzer et al., 2011), DTD (Cimpoi et al., 2014), Caltech101 (Fei-Fei et al., 2004), EuroSAT (Helber et al., 2018; 2019), PatchCamelyon (Veeling et al., 2018), RESISC45 (Cheng et al., 2017), and Rendered SST2 (Socher et al., 2013), which cover various visual domains. For OOD evaluation, we use the "real", "sketch", "inpainting", and "clippart" of DomainNet (Peng et al., 2019), where we train on either "real" or "sketch" and evaluate on the others. For CLIP pre-trained models, we additionally use 6 ImageNet variants (Recht et al., 2019; Hendrycks et al., 2021a; Wang et al., 2019a; Hendrycks et al., 2021b; Shankar et al., 2021; Barbu et al., 2019) for OOD evaluation while train on ImageNet-1K. We report the LP performance for both ID and OOD evaluation using $\{10\%, 25\%, 50\%, 75\%, 100\%\}$ percentage of downstream datasets. The setup can be extended to other architectures and pre-training proxy objectives, as shown in Section 4. Our pre-training primarily follows Wightman et al. (2021) and Cherti et al. (2023), with similar performance achieved as shown in Appendix A.1. More details of the setup are included in Appendix A.2.

## 2.2 RESULTS

In Figure 2, we plot the average accuracy for ID and OOD tasks of adapting the IN-1K fully supervised and YFCC15M CLIP pre-trained ResNet-50 models. With the extensive motivating experiments, we empirically find two important and counter-intuitive observations from the results:

---

[3]We always treat $y \in [C]$ as an actual class label on downstream datasets.

[4]Linear probing is an evaluation protocol accessing feature quality (He et al., 2019; Liu et al., 2021a).

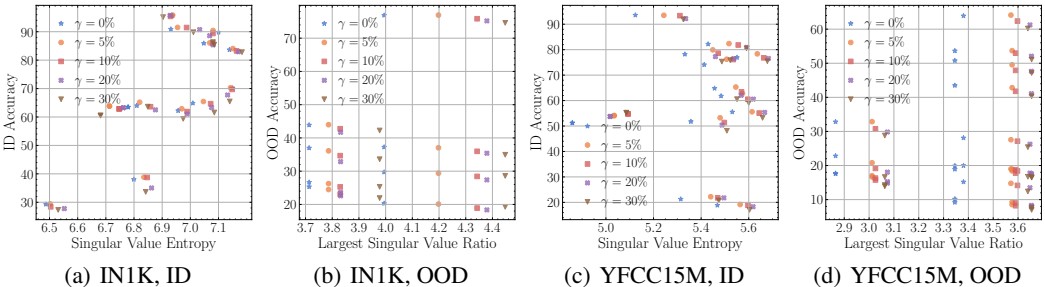

(a) IN1K, ID      (b) IN1K, OOD      (c) YFCC15M, ID      (d) YFCC15M, OOD

Figure 3: Feature SVD analysis. We compute the singular value entropy (SVE) for in-domain (ID) tasks and the largest singular value ratio (LSVR) for out-of-domain (OOD) tasks. Both metrics are computed for ImageNet-1K fully supervised pre-trained ((a) and (b)) and YFCC15M CLIP pre-trained ((c) and (d)) models. The SVE first slightly improves as the noise ratio increases to 5% or 10%, indicating better generalization. As the noise ratio increases, the SVE further improves, and the LSVR drops significantly, corresponding to worse generalization on ID and OOD tasks, as more noise structure is learned. The dominant singular components become less transferable.

- Proper noisy labels in pre-training (e.g., 5% or 10%) can benefit the performance on ID downstream tasks, while more noise results in inferior results;
- The robustness of transferability on OOD downstream tasks constantly deteriorates as the noise increases, even with the improvement in ID tasks on 5% noise.

While prior arts in noisy label learning mainly aim to correct/eliminate the noise or perform robust learning against noise (Ghosh et al., 2017; Li et al., 2020; Liu et al., 2022a; Xue et al., 2022), we show that the noise in pre-training can have both benevolent and malignant effects on downstream tasks. These observations raise a natural and fundamental question: *where does the superior transferability (with slight noise) and the inferior robustness stem from*? We further analyze the feature space to understand the change in the pre-trained feature extractor caused by noise.

## 2.3 FEATURE SPACE ANALYSIS

To understand the noise in pre-training data, we empirically analyze the singular value spectrum of the pre-trained feature space on downstream datasets, which is widely considered to be related to the generalization performance (Oymak et al., 2019; Chen et al., 2019; Xue et al., 2022). More specifically, we perform singular value decomposition (SVD) on the features $\mathbf{F} \in \mathbb{R}^{M \times D}$ [5] of pre-trained feature extractors on a downstream dataset: $\mathbf{F} = \mathbf{U}\mathbf{\Sigma}\mathbf{V}^\top$.[6] We plot the singular values in Appendix A.4, based on which we define two metrics that can help understand the observations:

**Definition 2.1** (Singular Value Entropy). The singular value entropy (SVE) is defined as the entropy of normalized singular values. SVE measures the flatness of the singular value distribution.

$$\text{SVE} = -\sum_{i=1}^{D} \frac{\sigma_i}{\sum_{j=1}^{D} \sigma_j} \log \frac{\sigma_i}{\sum_{j=1}^{D} \sigma_j} \tag{1}$$

Larger SVE values indicate that the feature space captures more structure in the data and thus spans more dimensions either due to more discriminated features are learned or memorization of the noise.

**Definition 2.2** (Largest Singular Value Ratio). The largest singular value ratio (LSVR) is defined as the logarithm of the ratio of the largest singular value $\sigma_1$ to the summation of all singular values:

$$\text{LSVR} = -\log \frac{\sigma_1}{\sum_{i=1}^{D} \sigma_i}. \tag{2}$$

LSVR measures the variations in data captured by the singular vector corresponding to the largest singular value $\sigma_1$, which relates to the transferability of a model (Chen et al., 2019).

---

[5] We denote $M$ as the number of samples in downstream datasets and $D$ as the feature dimension.

[6] We assume $D \leq M$ (Kumar et al., 2022). $\mathbf{U}$ and $\mathbf{V}$ denotes the left and right singular vector matrices, respectively, and $\mathbf{\Sigma}$ denoting the diagonal singular value matrix $\{\sigma_1, \ldots, \sigma_D\}$.

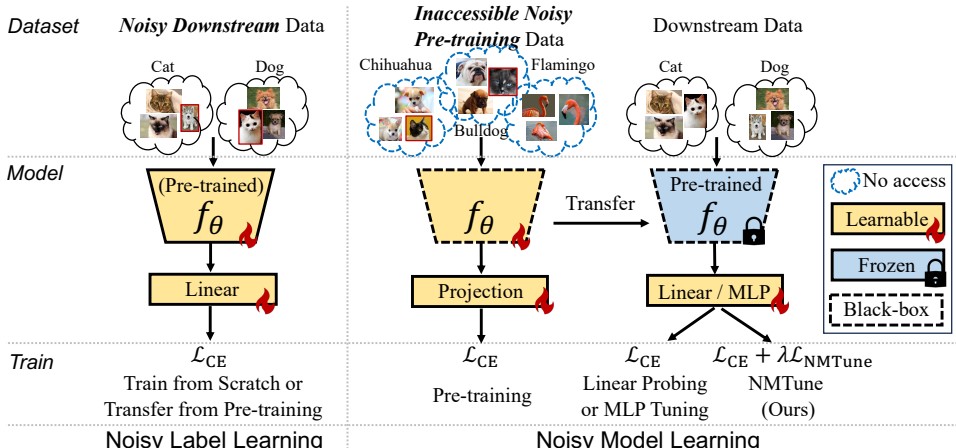

Figure 4: Illustration of noisy label learning (left) and the proposed *Noisy Model Learning* (right). Noisy label learning mainly focuses on robustly training a model from scratch or fine-tuning a model from pre-training on a noisy dataset. Noisy model learning focuses on robustly adapting the black-box noisy pre-trained models to downstream datasets with no assumption on the downstream dataset.

**Analysis.** We plot the SVE for ID tasks and LSVR for OOD tasks, as shown in Figure 3. For ID tasks, as the noise ratio slightly increases, the learned representation usually presents slightly higher SVE, which indicates the pre-trained feature extractor captures more structure in data. Specifically, more capabilities of the feature space are assigned to fit the noise in data, resulting in a feature space spanning more dimensions, which provides better-initialized features on downstream tasks and facilitates generalization. Similar observations have also been found and explored in Wu et al. (2022). However, as the noise ratio further increases, the increased SVE indicates that a more noisy data structure is captured and memorized, thus leading to deteriorated generalization performance. When the labels in pre-training are random, the SVE of the feature extractor would further increase by memorizing all the noise but not generalize on downstream tasks, similar to Zhang et al. (2021b). For OOD tasks, the robustness performance is *negatively correlated* with the LSVR. As the noise ratio increases, the LSVR consistently increases with the decreasing largest singular value. A less transferable component is learned, thus resulting in worse generalization on unseen OOD tasks.

## 3 MITIGATING THE NOISE WITH REGULARIZATION ON SINGULAR VALUES

In this section, we propose a black-box fine-tuning method, which we call "Noisy Model Tuning" (NMTune, Figure 4) in response to the noisy model learning setting. We demonstrate that NMTune can boost the generalization on downstream tasks and provide the analysis for the reasons behind.

### 3.1 METHOD

Per analysis above, noise in pre-training can shape the feature space differently from pre-training on clean data, reducing the top dominant singular values with dampened transferability while increasing the spanning dimensions of the feature space to fit noise structure. Since the large pre-trained models are usually difficult to fully fine-tune due to the enormous parameter size and limited computation resources, we propose to alter the pre-trained feature space $\mathcal{F}$ in a light-weight and black-box fashion. More specifically, we introduce a multi-layer perceptron (MLP) $h_\omega$ transforming the pre-trained features into new feature space $\mathcal{Z}$. We propose three regularization terms on $\mathbf{Z}$, to encourage the pre-trained knowledge to be maintained and improving SVE and LSVR of the new feature space.

**Consistency regularization**. To encourage the consistency of the pre-trained knowledge, we adopt a mean-square-error (MSE) loss between the normalized features $\mathbf{F}$ and $\mathbf{Z}$:

$$\mathcal{L}_{\mathrm{MSE}} = \left\| \frac{\mathbf{F}}{\|\mathbf{F}\|_2} - \frac{\mathbf{Z}}{\|\mathbf{Z}\|_2} \right\|_2^2. \tag{3}$$

This objective facilitates inheriting the pre-trained knowledge in the transformed features $\mathbf{Z}$.

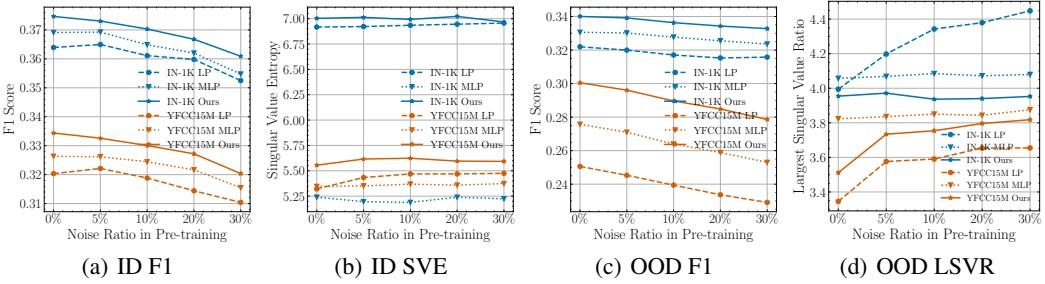

(a) ID F1      (b) ID SVE      (c) OOD F1      (d) OOD LSVR

Figure 5: Evaluation of our method on ID and OOD downstream tasks, compared to MLP tuning and LP on ResNet-50 models pre-trained on ImageNet-1K (IN-1K) and YFCC15M. (a) Average F1 score on ID tasks; (b) SVE on ID tasks; (c) Average F1 score on OOD tasks; (d) LSVR on OOD tasks. Our method presents better SVE and LSVR on both ID and OOD tasks with better generalization performance. Our method also rectifies the malignant noise effect: the feature extractor pre-trained on clean data now exhibits better performance than others on noisy data on ID tasks; and the performance gap between the clean one and the one with $5\%$ noise gets smaller on OOD tasks.

**Covariance regularization.** We define the covariance loss to encourage the off-diagonal elements in the covariance matrix of the transformed feature $C(\mathbf{Z})$ to be close to $\mathbf{0}$:

$$\mathcal{L}_{\text{COV}} = \frac{1}{D} \sum_{i \neq j} [C(\mathbf{Z})]_{i,j}^2, \text{ where } C(\mathbf{Z}) = \frac{1}{M-1} \sum_{i=1}^{M} (z_i - \bar{z})(z_i - \bar{z})^T, \bar{z} = \frac{1}{M} \sum_{i=1}^{M} z_i. \quad (4)$$

Inspired by Zbontar et al. (2021) and Bardes et al. (2022), we use the covariance regularization term to improve the SVE of feature space by preventing the different coordinates of the features from encoding similar information. It also encourages more discriminative features to be learned.

**Dominant singular value regularization**. To help transferability, we use a more specific regularization to improve the LSVR by directly maximizing the ratio of the largest singular value:

$$\mathcal{L}_{\text{SVD}} = -\frac{\sigma_1}{\sum_{j=1}^{D} \sigma_j}. \quad (5)$$

In summary, the total objective on a downstream task becomes:

$$\mathcal{L} = \mathcal{L}_{\text{CE}} + \lambda \mathcal{L}_{\text{NMTune}} = \mathcal{L}_{\text{CE}} + \lambda \left( \mathcal{L}_{\text{MSE}} + \mathcal{L}_{\text{COV}} + \mathcal{L}_{\text{SVD}} \right), \quad (6)$$

where $\mathcal{L}_{\text{CE}}$ is the cross-entropy loss for downstream classification. We set $\lambda = 0.01$ and use 2 layers MLP for all our experiments. Ablation study on MLP architecture and $\lambda$ are in Appendix B.7.

### 3.2 EVALUATION ON NOISY IMAGENET-1K AND YFCC15M

Here, we evaluate the proposed NMTune on the noisy models and analyze the reason for its effectiveness. We compare against solely training the MLP without the regularization, termed as MLP tuning, to show the effectiveness stems from the regularization rather than the extra parameters.

For ID tasks, we plot the average F1 score and SVE in Figures 5(a) and 5(b), respectively. The F1 score of linear probing (LP) on different pre-training noise ratios follows the same trend as the accuracy: it first increases as the noise ratio goes up to $5\%$ and then decreases. While adding an MLP can improve the F1 score in general, we find that it cannot mitigate the effect of noise, i.e., the clean pre-trained model underperforms the $5\%$ noisy pre-trained models. Further introducing our method can rectify the effect of noise on ID tasks, leading the clean pre-trained feature extractor to achieve the best results. More interestingly, only adding a MLP to LP can result in a smaller SVE, especially on ImageNet-1K, corresponding to a much sparser feature structure. In contrast, our method provides a larger and flatter SVE. It indicates the transformed feature space not only maintains the pre-trained knowledge but also spans more dimensions. For OOD tasks, the F1 score and LSVR are shown in Figure 5(c) and 5(d), respectively. Similarly, one can observe significantly better generalization performance deploying NMTune, compared to the MLP and LP. We also notice a smaller performance gap for the clean pre-trained feature extractor and $5\%$ noisy pre-trained,

Table 1: Results on popular vision models that are pre-trained on noisy datasets. We use 14 in-domain (ID) and 4 out-of-domain (OOD) tasks.

| Pre-trained Model | Tuning Method | In-Domain | | Out-of-Domain | |
|---|---|---|---|---|---|
| | | Acc. | F1 | Acc. | F1 |
| JFT300M | LP | 76.72 | 0.3815 | 44.13 | 0.3594 |
| Semi-Supervised | MLP | 76.87 | 0.3833 | 45.95 | 0.3624 |
| EfficientNet-B3 | Ours | **77.63** | **0.3874** | **46.84** | **0.3654** |
| ImageNet-21K | LP | 77.51 | 0.3718 | 40.82 | 0.3062 |
| Fully Supervised | MLP | 77.58 | 0.3726 | 41.73 | 0.3053 |
| ResNetv2-152x2 | Ours | **78.43** | **0.3862** | **42.42** | **0.3100** |
| ImageNet-21K | LP | 81.91 | 0.4092 | 50.88 | 0.3838 |
| Fully Supervised | MLP | 82.51 | 0.4128 | 51.21 | 0.3811 |
| Swin-L | Ours | **84.16** | **0.4177** | **52.35** | **0.3901** |
| Laion-2B | LP | 88.86 | 0.4432 | 66.86 | 0.4253 |
| CLIP | MLP | 88.53 | 0.4417 | 68.43 | 0.4304 |
| ConvNext-L | Ours | **89.48** | **0.4457** | **70.30** | **0.4367** |
| Laion-2B | LP | 86.85 | 0.4328 | 66.89 | 0.4208 |
| CLIP | MLP | 87.23 | 0.4375 | 69.50 | 0.4221 |
| ViT-L | Ours | **88.57** | **0.4414** | **70.47** | **0.4246** |

Table 2: Evaluation of our method on language models in practice that are pre-trained on noisy datasets. We use GLUE for in-domain (ID) tasks and GLUE-X for out-of-domain (OOD) tasks.

| Pre-trained Model | Tuning Method | In-Domain | Out-of-Domain |
|---|---|---|---|
| BERT-L | LP | 69.44 | 50.65 |
| | MLP | 69.78 | 50.62 |
| | Ours | **70.26** | **51.63** |
| RoBERTa-L | LP | 69.75 | 44.55 |
| | MLP | 70.27 | 45.22 |
| | Ours | **70.97** | **47.01** |
| GPT-2 | LP | 58.67 | 36.68 |
| | MLP | 58.44 | 37.24 |
| | Ours | **59.34** | **39.07** |
| text-ada-002 | LP | 56.96 | 44.06 |
| | MLP | 63.89 | 51.30 |
| | Ours | **65.99** | **53.48** |

especially on YFCC15M. On LSVR, MLP tuning usually imposes larger LSVR compared to LP, presenting smaller dominant singular values. Considering MLP tuning also presents smaller SVE, its resulting feature space is expected to present a more long-tailed spectrum than the original feature space. Maximizing the dominant singular values results in better transferability for OOD tasks.

# 4 EXPERIMENTS

We further validate NMTune on practical large-scale vision and language models that are pre-trained on noisy data, and discuss the noisy label learning and running time analysis in this section.

## 4.1 VISION MODELS AND DATASETS

**Setup**. For vision models, we use ResNet152 (He et al., 2016a) with dimensions widened by a factor of two (ResNet152x2) fully supervised pre-trained on ImageNet-21K (Kolesnikov et al., 2020), Swin-L (Liu et al., 2021c) fully supervised pre-trained on ImageNet-21K, EfficientNet-B3 semi-supervised pre-trained on noisy JFT-300M (Hinton et al., 2015; Chollet, 2017) and ImageNet-1K, and ViT-L (Dosovitskiy et al., 2020) and ConvNext-L (Liu et al., 2022c) contrastive pre-trained on noisy Laion-2B (Cherti et al., 2023). All pre-trained models are adapted from TIMM (Wightman, 2019). We evaluate the models on the 14 downstream ID and 4 OOD vision datasets as in Section 2. The details of hyper-parameters are shown in Appendix B.1 due to space limit.

**Results**. We present the average accuracy and F1 score across different datasets with three runs on vision models in Table 1. Our method improves the quality of the noisy pre-trained features with better accuracy and F1 score on both ID and OOD vision tasks. A large margin on downstream task across different pre-training architectures and datasets is present by NMTune, demonstrating better feature is learned. Noteworthy is that, although the MLP tuning also improves the performance in general, its performance gain is much smaller compared to our method, showing the effectiveness of the proposed regularization terms on mitigating the malicious effect of noise and improving generalization. More detailed results with error bars for each dataset are shown in Appendix B.2.

## 4.2 LANGUAGE MODELS AND DATASETS

**Setup**. We evaluate BERT-L (Devlin et al., 2018), RoBERTa-L (Liu et al., 2019), and GPT-2 (Radford et al., 2019) on the GLUE (Wang et al., 2018) and GLUE-X (Yang et al., 2023) benchmarks for ID and OOD performance.. BERT-L and RoBERTa-L are pre-trained on the combination of the BooksCorpus data (Zhu et al., 2015) and English Wikipedia with uncompressed raw text. It is found that the raw pre-training data of BERT can be reduced from 16GB to 12GB with data cleaning (Yang et al., 2019). GPT-2 is pre-trained on WebText (Radford et al., 2019), a scraped web dataset from Common Crawl that contains low-quality raw texts. We also leverage OpenAI's API service "text-ada-002"[7]. Details of the hyper-parameters and evaluation metrics are in Appendix B.3.

---

[7]We cannot use larger and more recent language models such as LLaMA (Touvron et al., 2023a), since they are unable to fit in a single V100 GPU and we are unsure whether GLUE is in their training data.

**Results**. In Table 2, NMTune consistently achieves the best generalization performance. It presents superior performance gain, especially on OOD tasks of GLUE-X. On the "text-ada-002" model with only API access, it also outperforms LP significantly, demonstrating the necessity of mitigating the effect of noise for better generalization. Interestingly, on the ID tasks of GLUE, we also observe a smaller gap of MLP tuning method to LP even with more parameters, showing that the MLP alone may not mitigate the influence of noisy data in pre-training. Full results are in Appendix B.4.

## 4.3 DISCUSSION

**Noisy model learning with noisy label learning**. We explore another setting, where these two paradigms occur together with both the pre-training and fine-tuning containing label noise, as shown in Appendix B.5. Our exploration in synthetic noisy CIFAR-10/100 presents similar observations of LP and NMtune as in clean downstream datasets, and they can work closely to achieve better performance on downstream datasets with slight noise. **Running time analysis**. We present the average GPU hours of NMTune, MLP tuning, and LP in Appendix B.6, showing that it introduces negligible computation. The ablation study and architecture of MLP are shown in Appendix B.7. Finally, our results may not be comparable to white-box full fine-tuning results, which is acceptable since we perform black-box tuning and the feature extractors are frozen. Our goal is not to pursue the best but to offer insights and discuss new research possibilities in the era of foundation models.

## 5 RELATED WORK

**Noisy label learning.** Prior arts on noisy label learning mainly focus on how to train robust models or how to adapt clean pre-trained models on noisy (downstream) datasets from scratch, including robust loss functions (Ghosh et al., 2017; Zhang & Sabuncu, 2018; Wang et al., 2019b; Ma et al., 2020), noise estimation (Xiao et al., 2015; Goldberger & Ben-Reuven, 2016; Liu et al., 2020; Northcutt et al., 2021; Li et al., 2021), and noise correction (Han et al., 2018; Li et al., 2020; Zhang et al., 2021c; Liu et al., 2022a; Kim et al., 2021; Chen et al., 2023). Perhaps more close to our work is the line of understanding noisy label learning. Ghosh et al. (2017) looked at theoretical conditions for a loss function to be noise-tolerant. CIFAR-N (Wei et al., 2022b) was built to understand the real-world instance-dependent label noise. Cheng et al. (2023) proposed to mitigate the memorization of noise labels by analyzing the regularization between representations. Wen et al. (2022) provably verified the failure of benign overfitting with label noise. Xue et al. (2022) investigated the robustness of contrastive pre-training with noisy labels on downstream tasks. Our work differs from the noisy label learning paradigm by focusing on the effect of pre-training noise on downstream.

**Pre-training and fine-tuning.** Pre-training and fine-tuning is the dominant transfer learning paradigm that allows a pre-trained model to adapt to a new, but similar, dataset. Many techniques are proposed for better transfer performance on the new dataset when it contains distribution shift (Cheng et al., 2023), unlabeled data (Sohn et al., 2020; Zhang et al., 2021a; Wang et al., 2023a), imbalanced data (Kang et al., 2019; Wang et al., 2023c), and noisy data (Wei et al., 2022a; Xue et al., 2022). There are also much relevant work studying and processing the pre-training data for better transfer performance by diversity trade-off (Kaplan et al., 2020; Zhang et al., 2023a), data selection (Entezari et al., 2023), quality-quantity trade-off (Magar & Schwartz, 2022; Nguyen et al., 2022; Lee et al., 2022; Carlini et al., 2022; Gadre et al., 2023), and specified fine-tuning methods (Tsai et al., 2020; Kumar et al., 2022; Wortsman et al., 2022; Goyal et al., 2023; Xu et al., 2023). Parameter-efficient transfer learning (He et al., 2021; Oh et al., 2023) is lightweight paradigms by adding adapters (Houlsby et al., 2019), low rank approximation (Hu et al., 2021), or prompt tuning (Liu et al., 2022b; 2021b). However, they all assume the availability of pre-trained models while we deal with black-box models. They also do not consider the noise in pre-training data.

## 6 CONCLUSION

We presented *Noisy Model Learning*, a new research direction for understanding and mitigating the effect of label noise in pre-training on downstream tasks. Extensive experiments demonstrate that proper noise in pre-training can benefit in-domain tasks and hurt out-of-domain tasks. We then proposed NMTune to mitigate the malignant effect of noise and improve the generalization performance of various noisy pre-trained models and APIs. While being the first study in this area, the explored models are still relatively small-scale in terms of pre-training, and we only use ResNet-50 for analytical experiments, due to the limited computing resources. We hope our work can inspire more researchers on this important and challenging topic in more practical settings.

ACKNOWLEDGMENT AND DISCLAIMER

Masashi Sugiyama was supported by the Institute for AI and Beyond, UTokyo. In this paper, we generated some noisy pre-training images using ImageNet-1K to thoroughly study the noisy pre-training data. Such noisy data indeed could have malignant influence on downstream tasks, according to our findings. The only purpose of conducting this research is to study the noisy pre-training data, but not to claim their instability in real applications. Additionally, all the generated noisy images and our pre-trained models based on these data are for research purpose only, and will be released per request.

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

# Appendix

## CONTENTS

## A   UNDERSTANDING THE NOISY LABELS IN PRE-TRAINING DATA

We provide additional experiment details for the motivating example of ResNet-50 in this section. We also present the detailed results on each downstream dataset for noisy pre-trained models on both ImageNet-1K and YFCC15M. The SVD plots on each dataset are also shown here.

### A.1   PRE-TRAINING DATASETS AND HYPER-PARAMETERS

For analysis in Section 2, we conduct pre-training of ResNet-50 on ImageNet-1K and YFCC15M.

For ImageNet-1K pre-training, we follow the training recipe in Wightman et al. (2021). To introduce noise in ImageNet-1K, we use function cleanlab (Northcutt et al., 2021) to introduce symmetric noise in each class. For YFCC15M CLIP pre-training, we follow the training recipe in Cherti et al. (2023). To introduce noise in YFCC15M, we swap the text description between two randomly sampled image-text pairs until the noise ratio is achieved. We show the validation accuracy on ImageNet-1K of the noisy ResNet-50 models pre-trained on ImageNet-1K and zero-shot accuracy on ImageNet-1K of the noisy ResNet-50 models pre-trained on YFCC15M in Table 3. The results show that our pre-training achieves the state-of-the-art results (Wightman et al., 2021; Cherti et al., 2023), as a basis for our further analysis.

### A.2   DOWNSTREAM VISION DATASETS AND HYPER-PARAMETERS

We present the details of the in-domain (ID) vision datasets in Table 4 and out-of-domain vision datasets Table 5. For ID, we conduct training on the training set and test on the validation set of the downstream dataset. For OOD on DomainNet (Peng et al., 2019), we conduct training on the training set of DomainNet Real or DomainNet Sketch, and test on all the other three DomainNet

Table 3: ImageNet-1K validation and zero-shot accuracy of ImageNet-1K pre-trained and YFCC15M CLIP pre-trained noisy ResNet-50 models.

| Noise Ratio | ImageNet-1K Pre-train Validation Accuracy | YFCC15M CLIP Pre-train Zero-shot Accuracy |
|---|---|---|
| 0% | 79.96 | 32.64 |
| 5% | 79.18 | 30.86 |
| 10% | 78.61 | 29.54 |
| 20% | 76.27 | 27.72 |
| 30% | 73.11 | 26.53 |

datasets not used in training. For OOD on ImageNet (Russakovsky et al., 2015), we conduct training on ImageNet training split and test on its variants.

To transfer a pre-trained model, we use linear probing (LP) for analysis as shown in Section 2. We train the linear classifier for 30 epochs on each downstream dataset, using AdamW (Kingma & Ba, 2014) optimizer with a cosine scheduler. We do not use weight decay for linear probing and set the learning rate to $0.1$ for all tasks.

Table 4: Details of the 14 in-domain (ID) vision datasets used to evaluate ID transfer performance of vision models.

| Dataset | Classes | Train Size | Test Size | Evaluation Metric |
|---|---|---|---|---|
| CIFAR-10 (Krizhevsky et al., 2009) | 10 | 50,000 | 10,000 | accuracy |
| CIFAR-100 (Krizhevsky et al., 2009) | 100 | 50,000 | 10,000 | accuracy |
| Flowers102 (Nilsback & Zisserman, 2008) | 102 | 2,040 | 6,149 | mean per class |
| Food101 (Fei-Fei et al., 2004) | 101 | 75,750 | 25,250 | accuracy |
| OxfordPet (Parkhi et al., 2012) | 37 | 3,680 | 3,669 | mean per class |
| StanfordCars (Krause et al., 2013) | 196 | 8,144 | 8,041 | accuracy |
| FGVCAircraft (Maji et al., 2013) | 102 | 6,667 | 3,333 | mean per class |
| SVHN (Netzer et al., 2011) | 10 | 73,257 | 26,032 | accuracy |
| DTD (Cimpoi et al., 2014) | 47 | 1,880 | 1,880 | accuracy |
| Caltech101 (Fei-Fei et al., 2004) | 102 | 3,060 | 6,084 | mean per class |
| EuroSAT (Helber et al., 2019) | 10 | 21,600 | 5,400 | accuracy |
| PatchCamelyon (Veeling et al., 2018) | 10 | 73,257 | 26,032 | accuracy |
| RESISC45 (Cheng et al., 2017) | 45 | 25,200 | 6,300 | accuracy |
| Rendered SST2 (Socher et al., 2013) | 2 | 6,920 | 1,821 | accuracy |

Table 5: Details of the 4 out-of-domain (OOD) DomainNet datasets and 6 out-of-domain (OOD) ImageNet variants used to evaluate OOD transfer performance of vision models.

| Dataset | Classes | Train Size | Test Size | Evaluation Metric |
|---|---|---|---|---|
| DomainNet Sketch (Peng et al., 2019) | 345 | 48,212 | 20,916 | accuracy |
| DomainNet Real (Peng et al., 2019) | 345 | 120,906 | 52,041 | accuracy |
| DomainNet Painting (Peng et al., 2019) | 345 | - | 21,850 | accuracy |
| DomainNet Clipart (Peng et al., 2019) | 345 | - | 14,604 | accuracy |
| ImageNet-V2 (Recht et al., 2019) | 1,000 | - | 10,000 | accuracy |
| ImageNet-R (Hendrycks et al., 2021a) | 200 | - | 30,000 | accuracy |
| ImageNet-Sketch (Wang et al., 2019a) | 1,000 | - | 50,889 | accuracy |
| ImageNet-A (Hendrycks et al., 2021a) | 200 | - | 7,500 | accuracy |
| ImageNet-ViD (Shankar et al., 2021) | 1,000 | - | | accuracy |
| ObjectNet (Barbu et al., 2019) | 112 | - | 18,574 | accuracy |

## A.3 DETAILED ID AND OOD LINEAR PROBING RESULTS

We present the detailed ID and OOD linear probing results we analyzed in Section 2 here.

The ImageNet-1K and YFCC15M pre-trained ID results are in Figure 6 and Figure 8 respectively. On all the datasets, we can observe that the $5\%$ or $10\%$ noise pre-trained models outperform the clean pre-trained models, no matter which pre-training dataset and method is used.

The OOD results are in Figure 7 and Figure 9 respectively. On the validation split of the training dataset (ID), the trend follows the ID observations, where 5% noisy pre-trained model is better. However, on the OOD datasets, the model performance deteriorates as noise increases.

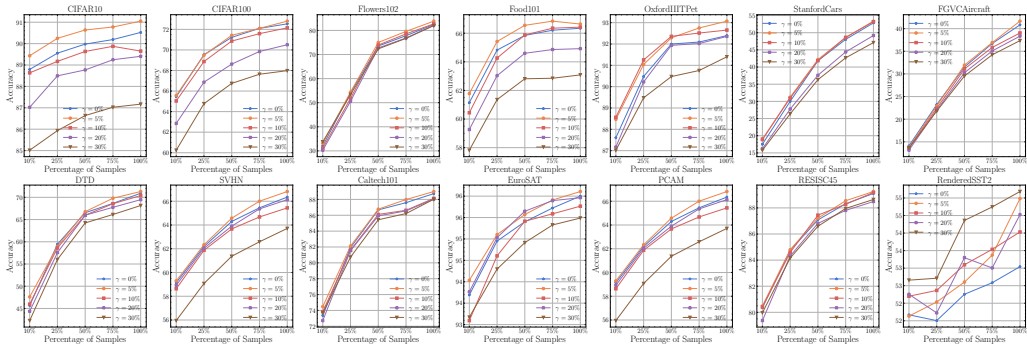

Figure 6: ImageNet-1K pre-trained ResNet-50 in-domain (ID) evaluation results

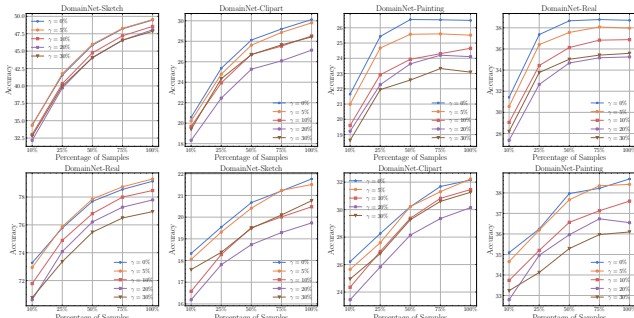

Figure 7: ImageNet-1K pre-trained ResNet-50 out-of-domain (OOD) evaluation results

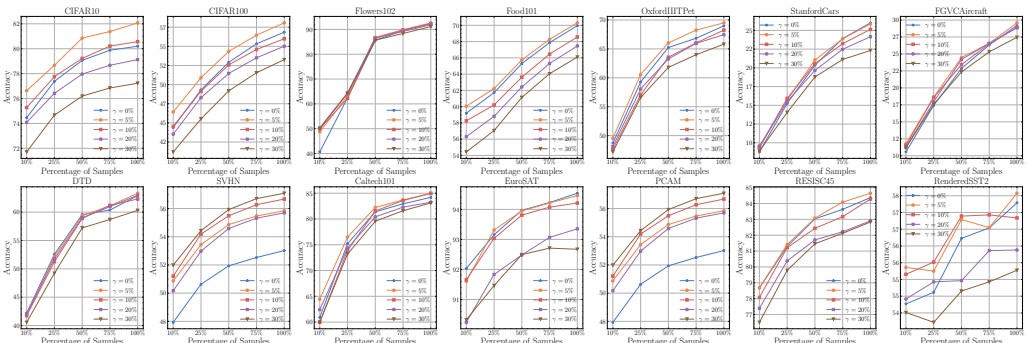

Figure 8: YFCC15M pre-trained ResNet-50 in-domain (ID) evaluation results

## A.4 DETAILED ID AND OOD SINGULAR VALUE SPECTRUM

We plot the singular value spectrum for ID datasets and OOD datasets of the noisy ResNet-50 models. To better visualize the spectrum, we split the singular values into three groups: the top 20, 20-500, and the remaining.

The singular value spectrum of the ID datasets is shown in Figure 11 and Figure 13 respectively. From 20-500 singular values visualization, we can observe that the noisy pre-trained models in general have larger singular values in this range, corresponding to a feature space that spans more of its coordinates. We summarize this visualization as the SVE introduced Section 2. Here, we provide

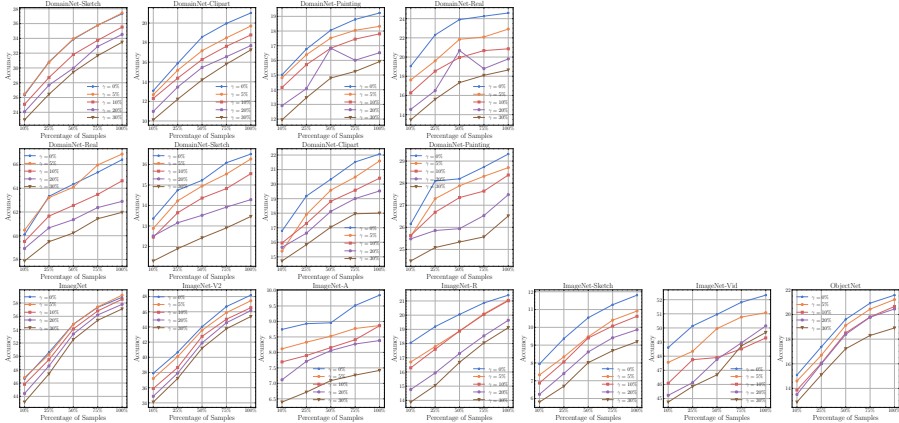

Figure 9: YFCC15M pre-trained ResNet-50 out-of-domain (OOD) evaluation results

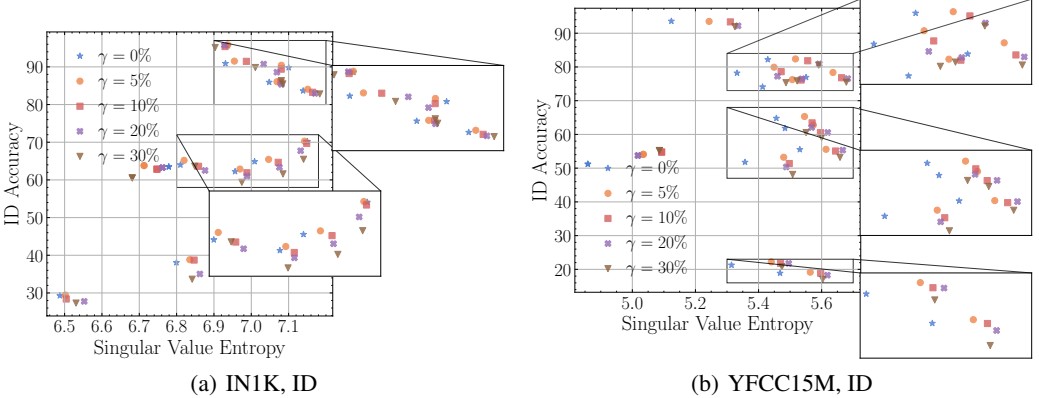

Figure 10: Zoom-in visualization of feature SVE analysis for in-domain (ID) tasks.

more explanation how to make Figure 3. First, each color and marker represents a different pre-training noise ratio. We plot the average accuracy of different percentage of downstream datasets and the SVD (or LSVR) of the downstream test data for each downstream task. Thus each points corresponds to a downstream task. The results of different pre-training noise ratio for each task are thus clustered together.

We also provide a zoom-in version for Figure 3(a) and Figure 3(b) for better visualization, as shown in Figure 10.

The singular value spectrum of the OOD datasets is shown in Figure 12 and Figure 14 respectively. From the top 20 singular values visualization, we can observe that the clean pre-trained model tends to present larger singular values in this range, especially the largest singular value. We connect this observation with the transferability performance on OOD tasks (Chen et al., 2019), and summarize it as LSVR introduced in Section 2.

# B  EXPERIMENTS

More details of experiments in Section 4 are shown here.

## B.1  DETAILED SETUP FOR VISION MODELS EXPERIMENTS

We provide a more detailed setup of evaluation on practical vision models. First, we summarize the noisy pre-trained models with their pre-trained dataset, parameter size, and validation accuracy on

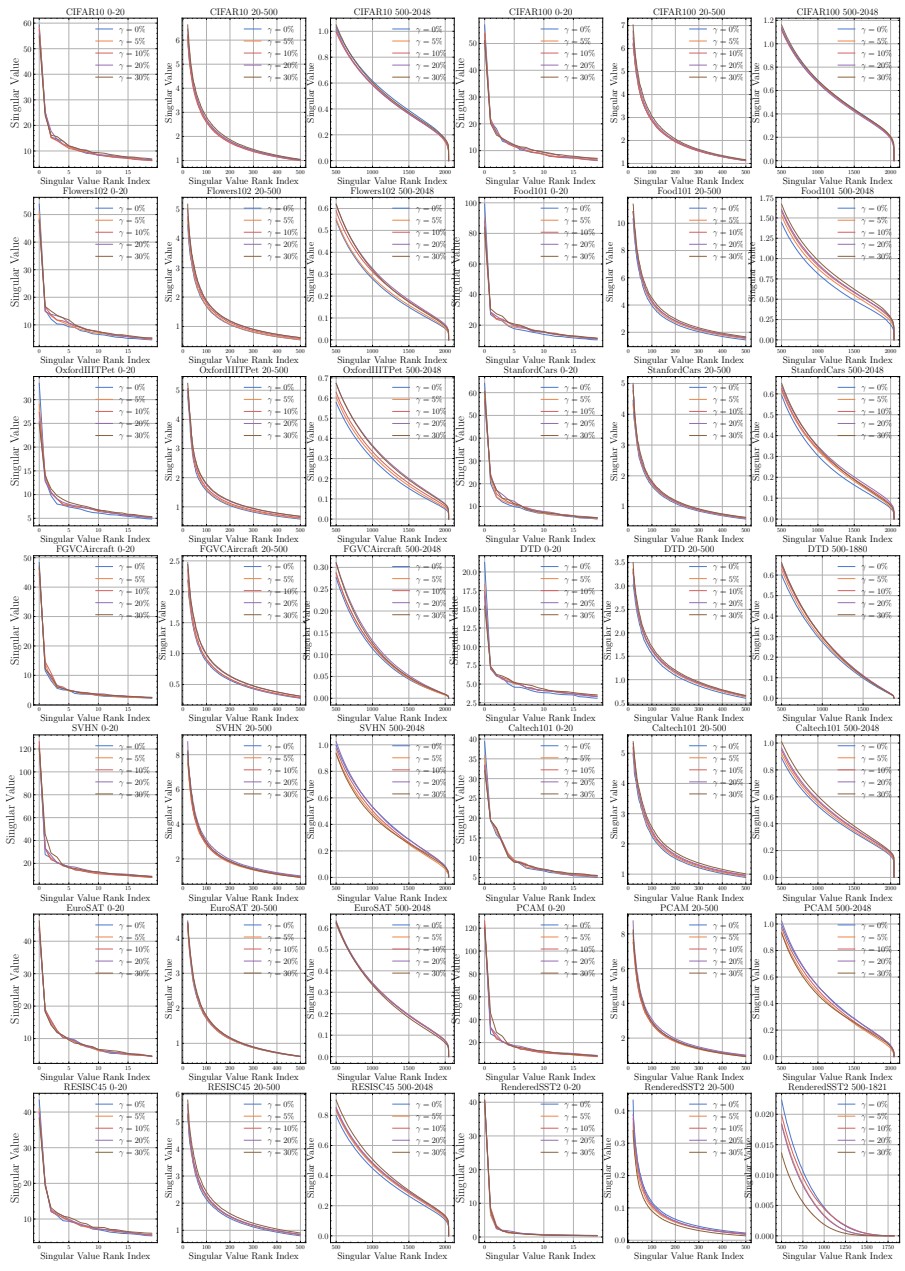

Figure 11: ImageNet-1K R50 in-domain (ID) feature SVD spectrum analysis

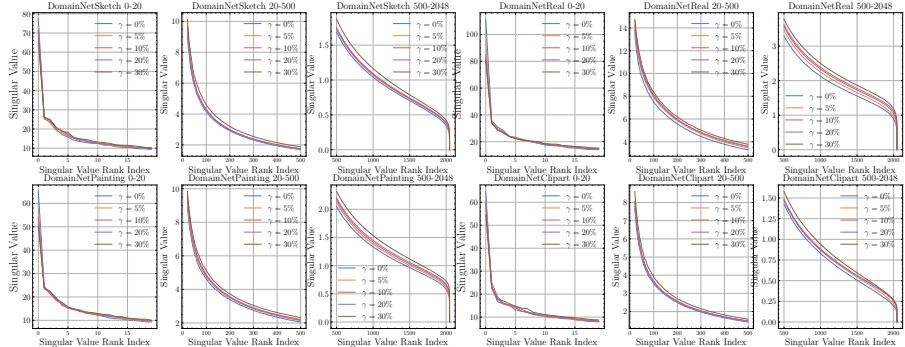

Figure 12: ImageNet-1K R50 out-of-domain (OOD) feature SVD spectrum analysis

ImageNet-1K we used in Table 6. We use the same ID vision and OOD vision datasets as in Table 4 and Table 5 for evaluation. Each experiment is run with three random seeds.

Table 6: Noisy vision models we evaluated.

| Model | Pre-trained Data | Pre-trained Method | Param. Size (M) |
|---|---|---|---|
| EfficientNet-B3 (Tan & Le, 2019) | JFT-300M (Hinton et al., 2015) | Noisy Student (Xie et al., 2020) | 12.23 |
| ResNetv2-152x2 (He et al., 2016b) | ImageNet-21K (Ridnik et al., 2021) | BiT Kolesnikov et al. (2020) | 236.34 |
| Swin-L (Liu et al., 2021c) | ImageNet-21K (Ridnik et al., 2021) | Supervised (Liu et al., 2021c) | 196.74 |
| ViT-L (Dosovitskiy et al., 2020) | Laion-2B (Schuhmann et al., 2022) | CLIP (Radford et al., 2021) | 304.20 |
| ConvNext-L (Liu et al., 2022c) | Laion-2B (Schuhmann et al., 2022) | CLIP (Radford et al., 2021) | 200.13 |

We mainly compare our method with MLP tuning and LP, where we fine-tuning the modules using AdamW (Kingma & Ba, 2014) for 30 epochs with a cosine learning rate scheduler. We set the learning rate as 0.1 and weight decay of 0 for LP, and set the learning rate as 0.001 and weight decay of $1e-4$ for MLP tuning and our method.

## B.2 DETAILED RESULTS FOR VISION MODELS EXPERIMENTS

More results on each evaluated dataset are provided here. The ID results with standard deviation in accuracy on each ID datasets are shown in Table 7, and the OOD results with standard deviation in accuracy on the evaluated OOD datasets are shown in Table 8.

Table 7: Evaluation of our method on vision models in practice that are pre-trained on noisy datasets. We compare different methods on 14 vision datasets for in-domain (ID) evaluation

| Pre-trained Model | Tuning Method | CIFAR10 | CIFAR100 | Flowers102 | Food101 | OxfordIIITPet | StanfordCars | FGVCAircraft | DTD | SVHN | Caltech101 | EuroSAT | PCAM | RESISC45 | RenderedSST2 | Avg |
|---|---|---|---|---|---|---|---|---|---|---|---|---|---|---|---|---|
| JFT-300M | LP | 94.68±0.12 | 79.00±0.23 | 91.43±0.09 | 79.71±1.02 | 94.92±0.11 | 59.36±0.98 | 43.82±1.34 | 73.60±0.20 | 63.77±0.99 | 90.65±0.32 | 95.88±0.01 | 63.77±1.44 | 89.62±0.87 | 53.93±0.23 | 76.72 |
| Semi-Supervised | MLP | 95.87±0.08 | 79.51±0.14 | 87.78±0.53 | 82.26±0.54 | 94.96±0.11 | 57.70±0.74 | 41.46±2.42 | 72.55±0.21 | 67.19±1.28 | 87.52±0.58 | 97.10±0.02 | 67.19±0.98 | 92.25±0.54 | 52.85±0.18 | 76.87 |
| EfficientNet-B3 | Ours | 96.15±0.05 | 79.71±0.10 | 91.61±0.09 | 82.63±0.51 | 95.29±0.13 | 60.24±0.69 | 43.57±1.02 | 73.78±0.18 | 67.05±0.54 | 88.73±0.44 | 97.20±0.01 | 67.05±0.91 | 92.60±0.40 | 51.25±0.08 | 77.63 |
| ImageNet-21K | LP | 96.39±0.13 | 85.21±0.17 | 96.85±0.08 | 85.99±0.47 | 91.73±0.20 | 55.96±1.23 | 43.37±0.98 | 73.51±0.55 | 62.51±0.83 | 92.15±0.16 | 97.15±0.13 | 58.70±0.87 | 92.20±0.43 | 53.47±0.32 | 77.51 |
| Fully Supervised | MLP | 97.02±0.11 | 85.47±0.19 | 96.96±0.09 | 86.46±0.34 | 92.39±0.23 | 57.25±1.12 | 41.10±1.45 | 73.76±0.43 | 64.61±0.56 | 89.50±0.21 | 97.59±0.11 | 59.66±0.45 | 93.33±0.27 | 51.02±0.59 | 77.58 |
| ResNetv2-152x2 | Ours | 97.12±0.11 | 85.68±0.16 | 96.98±0.08 | 85.69±0.35 | 92.49±0.18 | 57.42±0.99 | 43.54±1.23 | 73.87±0.46 | 66.94±0.50 | 92.01±0.12 | 97.67±0.09 | 61.12±0.41 | 93.97±0.35 | 53.53±0.71 | 78.43 |
| ImageNet-21K | LP | 98.06±0.07 | 88.51±0.12 | 99.27±0.04 | 89.68±0.63 | 92.60±0.19 | 65.43±0.34 | 53.48±1.44 | 77.45±0.39 | 72.94±0.33 | 90.91±0.09 | 97.00±0.50 | 72.94±0.23 | 93.84±0.21 | 54.59±0.70 | 81.91 |
| Fully Supervised | MLP | 98.45±0.06 | 89.72±0.09 | 99.12±0.05 | 91.20±0.30 | 93.75±0.24 | 68.15±1.22 | 52.81±0.78 | 77.63±0.42 | 73.48±0.32 | 89.26±0.22 | 98.07±0.10 | 74.48±0.37 | 95.04±0.23 | 54.09±0.52 | 82.52 |
| Swin-L | Ours | 98.60±0.06 | 90.14±0.11 | 99.53±0.03 | 91.40±0.33 | 94.22±0.22 | 75.60±2.12 | 55.95±0.91 | 79.97±0.20 | 76.61±0.12 | 90.36±0.26 | 98.06±0.11 | 76.61±0.41 | 95.34±0.23 | 55.92±0.48 | 84.17 |
| Laion-2B | LP | 98.15±0.08 | 88.83±0.24 | 98.72±0.12 | 91.68±0.28 | 94.07±0.15 | 94.68±0.33 | 65.94±0.45 | 83.88±0.04 | 82.41±0.21 | 96.37±0.03 | 97.78±0.01 | 82.41±0.21 | 95.44±0.05 | 73.70±0.10 | 88.86 |
| CLIP | MLP | 98.64±0.05 | 89.67±0.18 | 96.35±0.47 | 92.80±0.41 | 93.55±0.13 | 94.47±0.32 | 67.06±0.75 | 81.33±0.24 | 82.08±0.45 | 94.33±0.02 | 98.25±0.03 | 82.25±0.15 | 96.51±0.14 | 73.27±0.14 | 88.54 |
| ConvNext-L | Ours | 98.73±0.06 | 90.39±0.14 | 98.50±0.26 | 92.90±0.24 | 94.52±0.12 | 95.48±0.19 | 69.18±0.51 | 83.23±0.19 | 82.83±0.17 | 95.26±0.05 | 98.46±0.03 | 82.53±0.15 | 96.82±0.08 | 73.94±0.23 | 89.48 |
| Laion-2B | LP | 98.09±0.05 | 88.43±0.25 | 95.89±0.11 | 91.67±0.50 | 93.24±0.09 | 93.04±0.34 | 62.28±1.23 | 81.70±0.23 | 77.02±0.58 | 93.44±0.09 | 97.28±0.02 | 77.02±1.25 | 95.86±0.31 | 71.00±0.50 | 86.85 |
| CLIP | MLP | 97.52±0.14 | 88.33±0.34 | 95.54±0.40 | 92.12±0.36 | 93.41±0.09 | 93.34±0.31 | 63.11±0.84 | 81.97±0.15 | 77.81±0.64 | 92.35±0.15 | 97.38±0.01 | 79.11±0.35 | 96.54±0.05 | 72.74±0.51 | 87.23 |
| ViT-L | Ours | 98.65±0.05 | 88.96±0.22 | 98.67±0.15 | 92.78±0.27 | 94.51±0.10 | 94.65±0.52 | 67.29±0.47 | 82.98±0.14 | 79.25±0.72 | 93.65±0.08 | 98.59±0.03 | 79.35±0.30 | 97.19±0.19 | 73.51±0.43 | 88.57 |

## B.3 DETAILED SETUP FOR LANGUAGE MODELS EXPERIMENTS

The model details for natural language processing are shown in Table 9. We did not leverage larger language models mainly due to the limited computational resources. The recent open-sourced language models, such as Llama, have been trained on a very large-scale corpus of the web, evaluating them on GLUE and GLUE-X has the possibility to impose the problem of performing testing on the training samples.

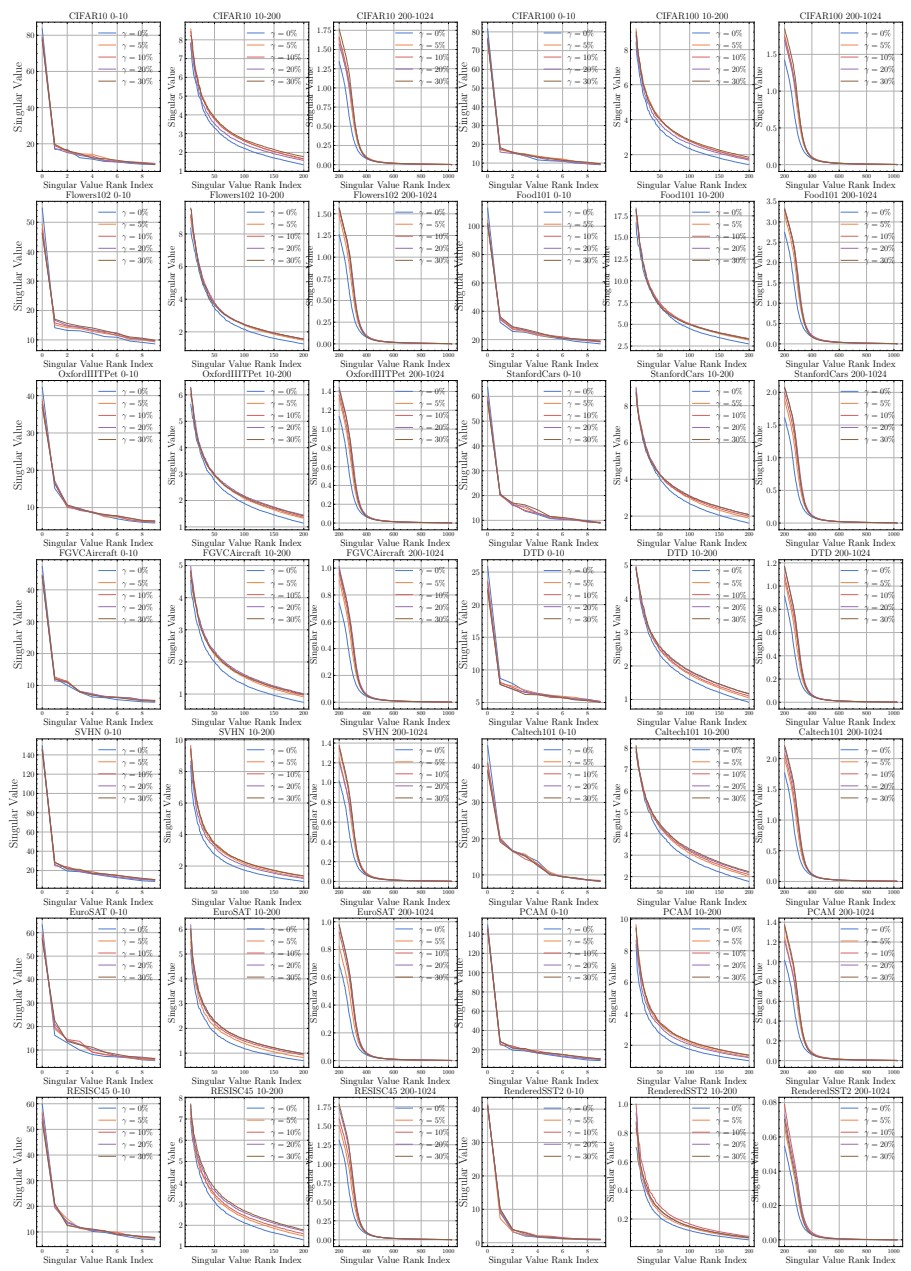

Figure 13: YFCC15M R50 in-domain (ID) feature SVD spectrum analysis

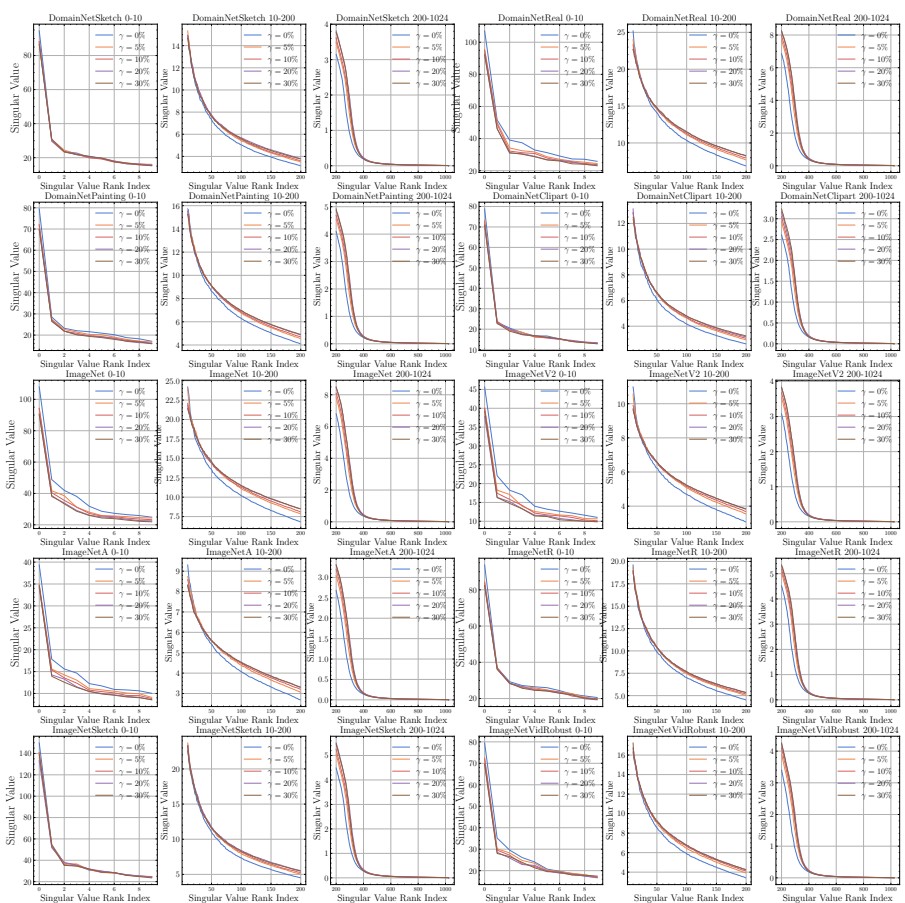

Figure 14: YFCC15M R50 out-of-domain (OOD) feature SVD spectrum analysis

Table 8: Evaluation of our method on vision models in practice that are pre-trained on noisy datasets. We compare different methods on 4 DomainNet datasets for out-of-domain (OOD) evaluation. We perform training on either DomainNetSketch or DomainNetReal, and evaluate on DomainNetSketch, DomainNetReal, DomainNetPaining, DomainNetClipart without the training set.

| Pre-trained Model | Tuning Method | DomainNet Sketch | DomainNet Real | Avg |
|---|---|---|---|---|
| JFT-300M | LP | 46.22±6.17 | 42.03±7.07 | 44.13 |
| Semi-Supervised | MLP | 48.29±5.78 | 43.61±7.53 | 45.95 |
| EfficientNet-B3 | Ours | **48.84±5.63** | **44.83±7.45** | **46.84** |
| ImageNet-21K | LP | 41.09±4.81 | 40.55±7.89 | 40.82 |
| Fully Supervised | MLP | 41.98±5.08 | 41.47±7.45 | 41.73 |
| ResNetv2-152x2 | Ours | **42.68±4.85** | **42.15±7.23** | **42.42** |
| ImageNet-21K | LP | 54.57±7.70 | 47.19±7.45 | 50.88 |
| Fully Supervised | MLP | 53.81±6.01 | 48.60±8.12 | 51.21 |
| Swin-L | Ours | **55.68±5.48** | **49.01±6.15** | **52.35** |
| Laion-2B | LP | 66.66±7.69 | 67.05±4.58 | 66.86 |
| CLIP | MLP | 66.78±7.00 | 70.07±4.90 | 68.43 |
| ConvNext-L | Ours | **69.74±6.82** | **70.85±4.77** | **70.30** |
| Laion-2B | LP | 67.00±6.94 | 66.77±4.74 | 66.89 |
| CLIP | MLP | 68.99±6.59 | 70.00±4.65 | 69.50 |
| ViT-L | Ours | **70.48±6.91** | **70.45±4.98** | **70.47** |

Table 9: Noisy language models we evaluated.

| Model | Pre-trained Data | Pre-trained Method |
|---|---|---|
| BERT-L (Devlin et al., 2018) | BooksCorpus and d English Wikipedia | Masked Modeling (Devlin et al., 2018) |
| RoBERTa-L (Liu et al., 2019) | BooksCorpus and d English Wikipedia | Masked Modeling (Liu et al., 2019) |
| GPT-2 (Radford et al., 2019) | WebText | Autoregression |
| text-ada-002 | - | - |

Now, we present the dataset details here used in our analysis. For ID evaluation, we use CoLA, SST-2, MRPC, STS-B, QQP, MNLI, QNLI, and RTE tasks of GLUE benchmark (Wang et al., 2018), as shown in Table 10. For OOD evaluation, following GLUE-X, we use Grammar Test (Yang et al., 2023) for CoLA, IMDB (Maas et al., 2011) for SST-2, QQP for MRPC, MNLI mismatched (Williams et al., 2017), SNLI (Bowman et al., 2015), SICK (Zhang et al., 2018) for MNLI, Reconstructed NewsQA (Trischler et al., 2016) for QNLI, SciTail (Khot et al., 2018) and HANS (McCoy et al., 2019) for RTE, as shown in Table 11.

Table 10: Details of the 8 in-domain (ID) tasks of GLUE used to evaluate ID transfer performance.

| Dataset | Classes | Train Size | Test Size | Evaluation Metric |
|---|---|---|---|---|
| CoLA | 2 | 8,500 | 1,000 | matthews correlation |
| SST-2 | 2 | 67,000 | 1,800 | accuracy |
| MRPC | 2 | 3,700 | 1,700 | accuracy |
| STS-B | 5 | 7,000 | 1,400 | pearson correlation |
| QQP | 2 | 364,000 | 391,000 | accuracy |
| MNLI | 3 | 393,000 | 20,000 | accuracy |
| QNLI | 2 | 105,000 | 5,400 | accuracy |
| RTE | 2 | 2,500 | 3,000 | accuracy |

We use the AdamW optimizer and set the learning rate for LP as 0.01 and for others as 0.001 for all the experiments of language models. For LP, we do not use weight decay, and for others we use a weight decay of 0.0001. All tuning methods are trained for 10 epochs with a linear learning rate scheduler.

## B.4 DETAILED RESULTS FOR LANGUAGE MODELS EXPERIMENTS

The detailed ID and OOD results of language models evaluation are shown in Table 12 and Table 13 respectively. NMTune outperforms LP and MLP tuning across all the tasks, whereas MLP tuning sometimes fall short than LP, demonstrating the necessitate of using the proposed regularization terms to help mitigate the effect of noise in pre-training and improve generalization performance.

Table 11: Details of the out-of-domain (OOD) tasks of GLUE-X used to evaluate OOD transfer performance.

| Dataset | Classes | Test Size | Evaluation Metric |
|---|---|---|---|
| Grammar Test | 2 | 304,277 | matthews correlation |
| IMDB | 2 | 50,000 | accuracy |
| MNLI mismatched | 2 | 9,832 | accuracy |
| SNLI | 2 | 570,152 | accuracy |
| SICK | 2 | 9,840 | accuracy |
| NewsQA | 2 | 119,525 | accuracy |
| SciTail | 2 | 26,527 | accuracy |
| HANs | 2 | 60,000 | accuracy |

Table 12: Evaluation of our method on language models in practice that are pre-trained on noisy datasets. We compare different methods on GLUE dev set for in-domain (ID) evaluation.

| Model | Tuning | CoLA MCC | MNLI Acc | MRPC Acc | QNLI Acc | QQP Acc | RTE Acc | SST2 Acc | STS PCC | Avg |
|---|---|---|---|---|---|---|---|---|---|---|
| BERT-L | LP | 46.18 | 62.75 | 72.05 | 74.10 | 84.83 | 50.90 | 88.19 | 76.55 | 69.44 |
| | MLP | **46.99** | 63.97 | 72.30 | 73.80 | 84.88 | 51.62 | 88.30 | 76.36 | 69.78 |
| | Ours | 46.12 | **64.37** | **73.04** | **74.25** | **85.11** | **52.54** | **88.84** | **76.64** | **70.26** |
| RoBERTa-L | LP | 41.09 | 59.27 | 76.47 | 76.07 | 83.19 | 55.96 | 90.02 | 75.93 | 69.76 |
| | MLP | 42.93 | 60.38 | 76.22 | 76.11 | 82.94 | 57.40 | 90.25 | 75.92 | 70.27 |
| | Ours | **43.91** | **60.59** | **77.69** | **76.33** | **83.30** | **58.96** | **90.71** | **76.24** | **70.97** |
| GPT-2 | LP | 4.86 | 53.11 | 72.55 | 67.65 | 78.82 | 56.68 | 84.75 | 40.93 | 57.42 |
| | MLP | 4.46 | 53.38 | 73.53 | 67.17 | 78.72 | 56.68 | 84.75 | 40.93 | 57.19 |
| | Ours | **6.41** | **54.21** | **73.77** | **68.09** | **78.95** | **57.07** | **85.14** | **41.09** | **58.09** |
| text-ada-002 | LP | 22.92 | 69.85 | 64.74 | 62.14 | 74.96 | 51.98 | 91.05 | 18.04 | 56.96 |
| | MLP | 33.90 | 70.49 | 66.42 | 69.54 | 84.28 | 52.34 | 91.85 | 42.28 | 63.89 |
| | Ours | **36.96** | **72.05** | **70.34** | **70.25** | **85.54** | **54.51** | **92.89** | **45.39** | **65.99** |

Table 13: Evaluation of our method on language models in practice that are pre-trained on noisy datasets. We compare different methods on GLUE-X for ouf-of-domain (OOD) evaluation.

| Model | Tuning | CoLA MCC | MNLI Acc | MRPC Acc | QNLI Acc | QQP Acc | RTE Acc | SST2 Acc | STS PCC | Avg |
|---|---|---|---|---|---|---|---|---|---|---|
| BERT-L | LP | 13.71 | 36.82 | 55.85 | 63.60 | 53.14 | 50.67 | 73.66 | 57.81 | 50.65 |
| | MLP | 14.11 | 36.49 | 56.15 | 63.61 | 52.34 | 50.02 | 73.32 | 58.91 | 50.62 |
| | Ours | **14.57** | **37.62** | **56.68** | **63.88** | **55.17** | **50.98** | **74.52** | **59.61** | **51.63** |
| RoBERTa-L | LP | 24.07 | 37.05 | 48.51 | 57.83 | 24.93 | 51.02 | 71.24 | 41.74 | 44.55 |
| | MLP | 24.50 | 37.32 | 48.96 | 58.56 | 24.12 | 52.51 | 73.86 | 41.95 | 45.22 |
| | Ours | **25.29** | **37.84** | **49.52** | **59.72** | **26.78** | **53.19** | **81.42** | **42.29** | **47.01** |
| GPT-2 | LP | 4.12 | 24.93 | 45.24 | 54.00 | 16.22 | 49.50 | 53.75 | 45.68 | 36.68 |
| | MLP | 4.89 | 25.20 | 47.99 | 54.11 | 17.21 | 49.71 | 53.60 | 45.21 | 37.24 |
| | Ours | **5.31** | **32.56** | **49.78** | **55.97** | **17.34** | **49.96** | **54.28** | **47.39** | **39.07** |
| text-ada-002 | LP | 8.06 | 65.14 | 38.66 | 58.12 | 32.60 | 51.53 | 84.07 | 14.30 | 44.06 |
| | MLP | 15.47 | 67.23 | 44.91 | 63.59 | 50.24 | 51.44 | 83.11 | 34.46 | 51.31 |
| | Ours | **17.50** | **68.97** | **46.41** | **63.81** | **55.34** | **52.81** | **85.16** | **37.80** | **53.48** |

## B.5 TRANSFERRING ON NOISY DOWNSTREAM DATASETS

We additionally study the setting where both pre-training and downstream datasets contain noisy labels. For pre-training noise, we use the ResNet-50 models pre-trained on noisy ImageNet-1K and YFCC15M with different noise ratios $\gamma \in \{0\%, 5\%, 10\%, 20\%, 30\%\}$, as in Section 2. For downstream noise, we adopt synthetic noise CIFAR-10 and CIFAR-100 which are usually used in noisy label learning (Liu et al., 2022a; Chen et al., 2023). We generate symmetric label noise by uniformly flipping labels for a percentage of the training set for all classes. We denote the noise ratio of downstream datasets as $\eta$, and set it to $\{0\%, 10\%, 20\%, 30\%, 40\%, 50\%\}$. We compare LP and NMTune in this setting, as shown in Figure 15 and Figure 16, respectively.

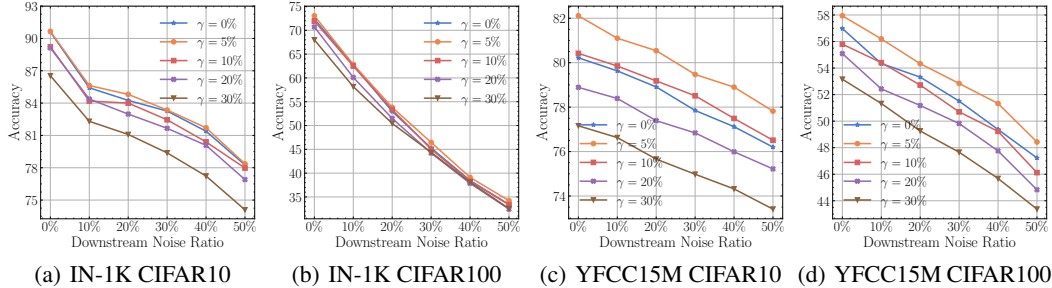

(a) IN-1K CIFAR10    (b) IN-1K CIFAR100    (c) YFCC15M CIFAR10    (d) YFCC15M CIFAR100

Figure 15: Linear Probing of noisy ResNet-50 models on noisy CIFAR-10 and CIFAR-100.

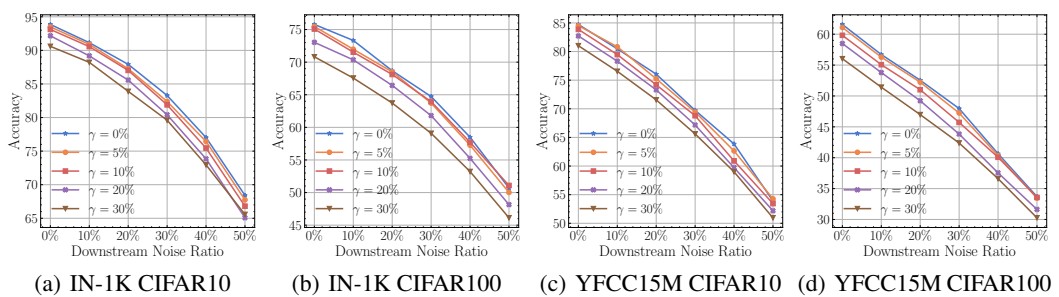

(a) IN-1K CIFAR10    (b) IN-1K CIFAR100    (c) YFCC15M CIFAR10    (d) YFCC15M CIFAR100

Figure 16: NMTune of noisy ResNet-50 models on noisy CIFAR-10 and CIFAR-100.

On the LP results in Figure 15, we find similar observations as our analysis in Section 2, where the 5% and 10% noisy pre-trained models usually outperforms the clean pre-trained model on downstream tasks, even the downstream tasks contain different level of noise. It indicates that the same conclusion from our main paper may extend and generalize to noisy downstream tasks, which highlights the importance of the proposed new topic - Noisy Model Learning - as the complementary to noisy label learning.

More importantly, we find that the proposed NMTune method has similar mitigation effect on noisy downstream tasks as the clean ones. On the NMTune results in Figure 16, we show that the clean pre-trained models now produce superior performance compared to noisy pre-trained models by utilizing the proposed regularization terms. It also improves the general performance when the noise ratio in downstream tasks is light, e.g., smaller than 40%. When the noise ratio in downstream tasks further increases, the performance of NMTune fall shorts to LP, which is acceptable because the regularization terms are not designed to be noise-tolerant. Noteworthy is that, even with slightly worse performance than LP, the performance of clean pre-trained mode still stays the best with NMTune. Devising NMTune to be more noise-tolerant on downstream tasks and experiments on practical asymmetric and instance-dependent noise (Wei et al., 2022b) would be very interesting and leave for the future exploration.

## B.6 RUNTIME ANALYSIS

The runtime analysis for NMTune, in comparison to LP and MLP tuning is shown in Table 14. All of our experiments on downstream are conducted on single NVIDIA V100 GPU. Thus we report the average GPU hours for running the ID and OOD evaluation of vision and language tasks. From the results, the proposed NMTune introduces minimal computation, compared to MLP with the exactly the same parameters. The additional computation burden may involve in the SVD calculation and the covariance matrix calculation on the features.

Table 14: Average run time for LP, MLP, and NMTune (Ours) in terms of GPU hours across in-domain and out-of-domain vision and language datasets.

| Datasets | LP | MLP | Ours |
|---|---|---|---|
| Vision In-Domain (14) | 5.01 | 7.96 | 8.23 |
| Vision Out-of-Domain (4) | 2.19 | 4.48 | 4.56 |
| Language In-Domain (8) | 2.87 | 3.76 | 3.82 |
| Language Out-of-Domain (14) | 1.21 | 1.34 | 1.45 |

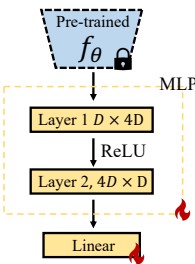

Figure 17: Architecture of the 2-layer MLP with ReLU activation.

### B.7 ABLATION STUDY

The ablation study of NMTune is present here, where we run evaluation on the ID vision datasets. We use three ResNet-50 models from ImageNet-1K pre-training and YFCC15M pre-training for ablation, including the clean pre-trained, $5\%$ noise pretrained, and $10\%$ noise pretrained.

We study the MLP architecture, more specifically, the non-linearity and the number of layers in MLP in Table 15. From the results, one can observe that removing the non-linearity reduces the performance significantly. Adding more layers only improves the performance slightly but introduces much more parameters. Thus we adopt the 2-layer MLP architecture with ReLU activation. The overall structure is shown in Figure 17.

Table 15: Ablation study of MLP architecture on ID vision datasets.

| MLP Layers | Activation | Models | IN-1K Pre-trained ID Accuracy | YFCC15M Pre-trained ID Accuracy |
|---|---|---|---|---|
| 2 | ReLU | 0 | 75.06 | 67.34 |
| | | 5 | 74.87 | 67.44 |
| | | 10 | 74.27 | 66.95 |
| 2 | None | 0 | 73.83 | 65.62 |
| | | 5 | 74.17 | 66.14 |
| | | 10 | 73.39 | 65.51 |
| 3 | ReLU | 0 | 75.16 | 67.56 |
| | | 5 | 74.99 | 67.49 |
| | | 10 | 74.48 | 67.13 |
| 4 | ReLU | 0 | 75.13 | 67.51 |
| | | 5 | 74.92 | 67.48 |
| | | 10 | 74.14 | 67.02 |

We also conduct ablation on the loss weight of different regularization terms we proposed in Table 16. From the results, we find that the proposed covariance regularization $\mathcal{L}_{\text{COV}}$ in general rectifies the effect of noise, improving the performance of clean pre-trained models to achieve better results than noisy pre-trained models. We can also observe that the dominant singular value regularization $\mathcal{L}_{\text{SVD}}$ helps improve generalization. Solely adding $\mathcal{L}_{\text{MSE}}$ or $\mathcal{L}_{\text{SVD}}$ does not produces this behavior and yields slight worse results.

## C MORE DISCUSSIONS

More discussions about our work are provided here.

Table 16: Ablation study of different loss weights on ID vision datasets

| $\mathcal{L}_{\text{MSE}}$ | $\mathcal{L}_{\text{COV}}$ | $\mathcal{L}_{\text{SVD}}$ | Models | IN-1K Pre-trained ID Accuracy | YFCC15M Pre-trained ID Accuracy |
|---|---|---|---|---|---|
| 0.01 | 0.01 | 0.01 | 0 | 75.06 | 67.34 |
| | | | 5 | 74.87 | 67.44 |
| | | | 10 | 74.27 | 66.95 |
| 0.00 | 0.01 | 0.01 | 0 | 74.34 | 66.59 |
| | | | 5 | 74.18 | 66.54 |
| | | | 10 | 74.09 | 66.17 |
| 0.01 | 0.00 | 0.01 | 0 | 73.65 | 65.41 |
| | | | 5 | 74.23 | 66.02 |
| | | | 10 | 73.56 | 65.39 |
| 0.01 | 0.00 | 0.00 | 0 | 74.16 | 66.27 |
| | | | 5 | 74.24 | 66.31 |
| | | | 10 | 73.17 | 66.08 |
| 0.01 | 0.01 | 0.00 | 0 | 74.74 | 67.03 |
| | | | 5 | 74.32 | 66.83 |
| | | | 10 | 73.92 | 66.70 |
| 0.00 | 0.01 | 0.00 | 0 | 74.41 | 66.51 |
| | | | 5 | 74.20 | 66.47 |
| | | | 10 | 73.98 | 66.12 |
| 0.00 | 0.00 | 0.01 | 0 | 72.21 | 65.08 |
| | | | 5 | 73.49 | 65.24 |
| | | | 10 | 72.87 | 64.76 |
| 0.00 | 0.00 | 0.00 | 0 | 73.96 | 66.11 |
| | | | 5 | 73.97 | 66.19 |
| | | | 10 | 73.13 | 65.93 |

## C.1 LIMITATIONS

The limitation mainly lies in our empirical study of the noise in pre-training. Due to the limited computing resources, we could only conduct experiments on reltively small scale backbone and datasets, while most of the SOTA foundation models are of much more parameters and are trained on much larger datasets. Also, the empirical experiments is limited to actual supervised pre-training. Other pre-training objectives will be explored in our future work. That being said, we do believe the observation and conclusions from our practical experiments could scale and extend to larger datasets, stronger backbones, and other training objectives.

## C.2 POTENTIAL FAILURE

We do observe some failure cases of the proposed methods. For example, from the results in Table.7, the proposed method falles short to LP on Caltech101 on almost all backbones we studied, while improving over MLP. Our hypothesis for the failure is that the SVD regularization term in the proposed method might need to optimize the top-K singular values instead of just the largest one. The optimal value of K might be different dataset. However, setting $K = 1$ can already achieves reasonable performance for most of the tasks.

