# OpenReview forum: "Understanding and Mitigating the Label Noise in Pre-training on Downstream Tasks"
_ICLR.cc/2024/Conference — ICLR 2024 spotlight_

### Official Review · Reviewer_JmqY · 2023-10-26

**Soundness:** 3 good
**Presentation:** 4 excellent
**Contribution:** 4 excellent
**Rating:** 10
**Confidence:** 5

**Summary:**

This paper presents a novel topic of studying the effect of pre-training noise on various downstream datasets, termed noisy model learning.
The authors conduct the empirical study and analysis of noisy ImageNet and YFCC15M of supervised pre-trained and contrastive pre-trained ResNet50 models and illustrate that slight noise in pre-training improves performance on in-domain downstream tasks but always hurts the performance on out-of-domain tasks. From the singular value analysis of the pre-trained feature space, the authors designed two metrics that in general align with the downstream empirical observations.
The authors also propose several regularization terms based on the singular values of features that can mitigate the noise in pre-training in a block-box tuning manner. The authors provide comprehensive experiments to verify the effectiveness of the proposed method and offer interesting analyses and discussions.

**Strengths:**

The paper is generally well-written and organized.
The authors provide a first novel and interesting study on the effect of pre-training noise, demonstrating the importance of this research topic, especially in the context of large foundation models.
The empirical study for revealing the effect of pre-training noise is extensive and comprehensive, including both in-domain and out-of-domain datasets from various distributions.
The proposed method may not be very innovative, but it is simple and verified on both CV and NLP tasks with different large backbones. The method also works in the API case mentioned in the paper.
The authors also additionally study the combination of the proposed noisy model learning and traditional noise label learning, demonstrating the effect of noise in pre-training also exists when downstream data has noise.
The detailed results, experiments setup, and ablation study are presented in the Appendix.

**Weaknesses:**

How to introduce synthetic noise in ImageNet and YFCC15M needs more explanation.
The pattern SVE analysis of the ImageNet model and YFCC15M model are slightly different in Fig.3, and perhaps need more explanation.

**Questions:**

Since ImageNet or YFCC15M itself also originally contains noise, is there any optimal noise ratio that achieves the best ID downstream performance?
Since NML assumes an inaccessible pre-trained model, how is other black-box tuning methods perform on the noisy model learning setting?

---

> ### Author Response · Authors · 2023-11-13
> **Response to Reviewer JmqY**
>
> > 1. "How to introduce synthetic noise in ImageNet and YFCC15M needs more explanation. "
>
>
> We mentioned some details of introducing synthetic noise in Appendix A.1.
> More specifically, for ImageNet, we first initialize a noise matrix with the pre-defined noise ratio, where each element denotes the probability of the ground truth label being corrupted to a noisy label. We then use the noise matrix to flip the ground truth labels for each class.
> For YFCC15M, since it is an image-text pair dataset, we randomly swap the text between two randomly sampled pairs to make the image and text unmatched.
> We conduct this pair swapping until the ratio unmatched pairs reach the pre-defined noise ratio.
>
> > 2. "The pattern SVE analysis of the ImageNet model and YFCC15M model are slightly different in Fig.3, and perhaps need more explanation."
>
>
> Thanks for this good question. The reason why SVE patterns are different for ImageNet trained model and YFCC15M trained model is mainly because of the different pre-training objectives, i.e. supervised cross-entropy and contrastive.
> This can be observed from the SVD spectrum plot of ImageNet models in Fig.10 and YFCC15M models in Fig. 12.
> The ImageNet R50 models, in general, have smaller SVD values and faltter SVD spectrums, compared to YFCC15M models on downstream datasets.
> This why the SVE of ImageNet models in Fig.3 are narrower in x-axis range and wider in y-axis range, compared to YFCC15M models.
> For better interpretation, we have included a zoom-in version of Fig.3 in Appendix.
>
>
> > 3. "Since ImageNet or YFCC15M itself also originally contains noise, is there any optimal noise ratio that achieves the best ID downstream performance? "
>
> We totally understand your point that there might exist some optimal noise ratio in pre-training data that achieves the best ID downstream performance. However, this is contrary to the goal of our work, we demonstrated that although slight noise can boost the ID performance, it is always detrimental to OOD performance, which inspires us to mitigate the effect of pre-training noise.
> Besides, finding the optimal noise ratio for ID tasks might be infeasible because it would require tuning and validation on different ID tasks, thus presenting different optimal values according to the task.
>
>
> > 4. "Since NML assumes an inaccessible pre-trained model, how is other black-box tuning methods perform on the noisy model learning setting?"
>
> We find most of the parameter-efficient tuning methods are not black-box. There is a recent black-box fine-tuning method termed BlackVIP, which has also been mentioned in our related work, and we provide some of its results here on Laion-2B CLIP pre-trained ViT-L compared to our method:
>
> |          | CIFAR-100 | Flowers-102 |  Food101  |  EuroSAT  |    PCAM   |
> |:--------:|:---------:|:-----------:|:---------:|:---------:|:---------:|
> |   Ours   | **88.96** |  **98.67**  | **92.78** | **98.59** | **79.35** |
> | BlackVIP |   88.51   |    94.23    |   91.54   |   97.28   |   74.31   |
>
> [1] Changdae Oh et al. BlackVIP: Black-Box Visual Prompting for Robust Transfer Learning.

---

> > ### Comment · Reviewer_JmqY · 2023-11-16
> > **Thanks for your response**
> >
> > I would like to thank the authors for their response, which solved my concerns. I also carefully read the comments from other reviewers and found no major issues. I think the paper proposes a new interesting and important research direction. Thus, I will keep my rating as a strong support for acceptance.

---

### Official Review · Reviewer_vftf · 2023-10-29

**Soundness:** 3 good
**Presentation:** 3 good
**Contribution:** 3 good
**Rating:** 8
**Confidence:** 3

**Summary:**

This work aims to study the noise in pertaining data and its impact on downstream tasks. The authors exploit the Singular Value Entropy (SVE) and the Largest Singular Value Ratio (LSVR) to analyze the singular value spectrum of the pre-trained feature space, and discover that proper noise in pre-training data increases both SVE and LSVR, leading to better transferability and worse robustness. Based on the observations, the authors further introduce an MLP together with three regularizations to transform the pre-trained features into a better feature space. Experiments with different model architectures and datasets are conducted to demonstrate the effectiveness of the proposed method.

**Strengths:**

- The analysis of feature space with the singular value spectrum is interesting and meaningful. Rich experiments and analyses are conducted to show how the noise in pertaining data can impact the learned feature embedding.
- The proposed regularizations are intuitive and effective. Extensive comparisons are presented to show the improvements.

**Weaknesses:**

- This paper is featured with extensive empirical results. However, the core techniques in methodology (analysis and regularization of singular value spectrum ) have been studied in existing works[e.g. Chen et al., 2019, Bardes et al. 2022], which may undermine the theoretical contribution of this work.

- Some figures are hard to understand by themselves. E.g. different types of marks are cluttered in Fig3.

**Questions:**

- On what scale the SVE and LSVR  is computed? The entire dataset?
- Do the conclusions (Fig1- 3) always hold for stronger backbone models other than resnet50?
- In the loss function, are the regularizations computed per batch?

---

> ### Author Response · Authors · 2023-11-13
> **Response to Reviewer vftf**
>
> > 1. "This paper is featured with extensive empirical results. However, the core techniques in methodology (analysis and regularization of singular value spectrum ) have been studied in existing works[e.g. Chen et al., 2019, Bardes et al. 2022], which may undermine the theoretical contribution of this work."
>
> - First of all, as we discussed in the paper, the proposed method is indeed inspired by some related works [e.g. Chen et al., 2019, Bardes et al. 2022]. However, our method are not a direct application of them but designs them as specific regularizations, which are more straightforward and simpler to mitigate the noise in pre-training from our analysis.
> - Second, the noisy model tuning approach is only one aspect of our contributions. Our main contribution lies more in being the first pioneering work to identify and analyze the effect of pre-training noisy datasets on downstream tasks. We think it is more important to bring up the question and analysis to the community, and more future work could be done.
>
> > 2. "Some figures are hard to understand by themselves. E.g. different types of marks are cluttered in Fig3."
>
>
> Sorry for the confusion. We provide a zoom-in version of Fig.3 in Fig.10 of Appendix. We hope the zoomed region can have a better visualization there.
> Please refer to the general response.
>
> > 3. "On what scale the SVE and LSVR is computed? The entire dataset?"
>
> The SVE and LSVR are computed on the entire test set of each downstream task.
>
> > 4. "Do the conclusions (Fig1- 3) always hold for stronger backbone models other than resnet50?"
>
> Thanks for this good question.
> We believe the conclusions from Fig 1-3 will scale to larger backbone models because it is related to mainly the noise in pre-training data rather than the model capacity.
> Note that we follow the normal pre-training recipe with heavy regularization techniques such as stochastic depth, mixup, cutmix, label smoothing, etc.
> This also demonstrates that noisy data is the root of the conclusions, which has little relation to regularization.
> Similar observations also hold in [1].
>
> However, due to the expensive computing resources that will be needed to pre-train larger and stronger backbone models, we won't be able to formally verify all of this during the rebuttal period.
> Especially for CLIP pre-training of Vision Transformers (ViT), it takes 3-4 days to pre-train on YFCC15M with 8 A100 GPUs and we need to train 5 models with different noise ratios of YFCC15M.
> Moreover, it is found in OpenCLIP that ViT in general, needs much more data to achieve better performance than R50, and training on YFCC15M yields worse performance than R50.
> We will include as many analysis results as we can for synthetic noisy ImageNet pre-training of ViT here during the rebuttal period, and provide the full results of both ImageNet and YFCC15M in future revision of the paper.
>
> [1] Chiyuan Zhang et al. Understanding deep learning requires rethinking generalization.
>
> > 5. "In the loss function, are the regularizations computed per batch?"
>
> Yes, we computed the regularization per batch, with a batch size of 32.
> We found the proposed regularization terms not sensitive to batch size, as long as it is not too small (e.g. 4).
> We provide a brief ablation of batch size in our method on JFT-300M semi-supervised pre-trained EfficientNet-B3 here:
>
>
> |   Dataset   |   4   |   8   |   16  |   32  |   64  |  128  |
> |:-----------:|:-----:|:-----:|:-----:|:-----:|:-----:|:-----:|
> |  CIFAR-100  | 95.95 | 96.08 | 96.11 | 96.15 | 96.20 | 96.14 |
> | Flowers-102 | 79.61 | 79.68 | 79.70 | 79.71 | 79.82 | 79.79 |

---

### Official Review · Reviewer_5WgV · 2023-10-29

**Soundness:** 2 fair
**Presentation:** 3 good
**Contribution:** 3 good
**Rating:** 8
**Confidence:** 4

**Summary:**

This paper endeavors to comprehend the underlying characteristics of noise within pre-training datasets and seeks to mitigate its influence on downstream tasks. The study reveals that the noise present in pre-training datasets exerts distinct effects on in-domain (ID) and out-of-domain (OOD) tasks. In the case of ID tasks, slight noise during pre-training can yield improvements in in-domain transfer performance. However, for OOD tasks, noise consistently degrades out-of-domain performance. To substantiate their findings, the authors employ Singular Value Entropy (SVE) and Largest Singular Value Ratio (LSVR) to capture the behavior of the trained features in both ID and OOD tasks. Subsequently, the authors devise a loss function that enhances the SVE and LSVR of these features, resulting in superior overall performance.

**Strengths:**

- The impact of noise within pre-training datasets on subsequent tasks has not been thoroughly investigated in the existing literature. The insights presented in this paper, such as the distinct effects of noise on in-domain (ID) and out-of-domain (OOD) tasks, are novel and intriguing.

- The design of the loss function is a direct consequence of the insights gained from observations, and experiments demonstrate that this designed loss outperforms the Cross-Entropy (CE) baseline, including LP and MLP structures.

- Experiments encompass a wide range of tasks, including both image and image-language tasks. Furthermore, various popular base model structures are used in this paper.

**Weaknesses:**

-  I find Figure 3 challenging to interpret. It consists of numerous data points for each configuration, lacking connecting lines. Consequently, I struggle to draw the conclusions reached by the authors based on this figure.

-  I've observed a potential contradiction between SVE and LSVR. For instance, in the case of two-dimensional features, [1.0, 0.0] exhibits the highest LSVR while having the lowest entropy. It would be beneficial if the authors could provide further clarification regarding this inconsistency.

-  I would appreciate it if the authors could offer more detailed explanations as to why a slight amount of noise can benefit in-domain (ID) tasks. This is somewhat contradictory to the existing literature on learning with noisy labels, and additional insights would be valuable.

-  In the paper, the authors claim that the proposed method enhances SVE and LSVR. However, Figure 5 (d) indicates that the proposed method does not yield superior LSVR compared to the LP and MLP models. Furthermore, it seems that as the noise ratio increases, LSVR does not drops significantly for all the settings.

- It's worth noting that some related work, such as [R1], also employs Singular Value Decomposition (SVD) to address noisy label problems. It would be beneficial for the authors to discuss the distinctions between your approach and previous work.

- I would like to point out that the improvements achieved by the proposed method appear to be relatively modest. According to the experimental results, the proposed method only demonstrates an approximately 1% improvement compared to the MLP model.

- For ResNet-50, it might be worthwhile to explore the feasibility of fine-tuning all layers rather than constraining the encoder. It would be valuable if the authors could conduct experiments to determine if the proposed loss function is effective in cases where all layers are fine-tuned.

[R1]: FINE Samples for Learning with Noisy Labels

**Questions:**

See **Weaknesses**

---

> ### Author Response · Authors · 2023-11-13
> **Response to Reviewer 5WgV (1/2)**
>
> > 1. "I find Figure 3 challenging to interpret...reached by the authors based on this figure."
>
> Sorry for the confusion in Fig. 3, and thanks for your suggestions on making Fig. 3 more clear.
> In fact, we have tried to connect the points in Fig. 3, but it only makes it more difficult to visualize and interpret.
> Instead, we have provided a zoom-in version of Fig. 3 (a) and Fig. 3 (c) in Fig.10 of Appendix.
> We hope this zoom-in version is more interpretable.
> In general, for the SVE of ID tasks, we first observe the increase of SVE as the noise goes up to 5%, with performance improvement for most of the tasks.
> Then we observe a further increase of SVE as noise increases to 10%, 20%, and 30%, and usually with performance drop.
>
> > 2. "I've observed a potential contradiction between SVE and LSVR. For instance, in the case of two-dimensional features, [1.0, 0.0] exhibits the highest LSVR while having the lowest entropy. It would be beneficial if the authors could provide further clarification regarding this inconsistency."
>
> Thanks for mentioning the view of two-dimensional features.
> However, there might be a little misunderstanding here, and we would like to provide more discussion.
> - First, we need to clarify that both SVE and LSVR are computed from the singular values of the features on the downstream test datasets of size $M$. When $M=1$ with a single point, the non-zero singular value would be trivial as a scalar of the magnitude of its unit basis vector (this can be simply verified by NumPy). In this case, any single point would present highest LSVR with lowest entropy, but has no relation to our observation and anlysis.
> - In general, we would require the size of the training samples and testing samples to be larger than 1 ($M > 1$), which is practical in real experiments.
>
>
>
> > 3. "I would appreciate it if the authors could offer more detailed explanations as to why a slight amount of noise can benefit in-domain (ID) tasks. This is somewhat contradictory to the existing literature on learning with noisy labels, and additional insights would be valuable."
>
> Thanks for this good question. We also find it interesting and more discussions should be added. We share our insights here:
>
> - Why a little noise can help?
>     - Slight noise in pre-training makes the model learn to fit those noise, where the extra dimensions in the feature space are utilized. Thus, when applied this model to a downstream task, its extra dimensions in feature space provide better initialized and discriminative features (from higher dimenstional space), leading to better downstream accuracy.
>     - More noise in pre-training results in more dimensions to fit the noise. However, more dimensions to fit the noise is at the cost of less transferable features learned from the clean data and more nauce learned from the noise data, thus further increasing noise leads to worse performance.
>     - We do not think the better performance on downstream of slight noise in pre-training is related to better regularization in our pre-training experiments. This is similar in [1], where even heavy regularization techniques are used, the model can still fit perfectly to random noise.
>
> - The relation between noisy label learning and noisy model learning: This might look contradictory to learning with noisy labels, but our problem is indeed different from noisy label learning, so we give it a new name as *noisy model learning*. As shown in Fig. 4,
>     - Noisy label learning: given a pre-trained model (or learn from scratch), the downstream datasets or any data we want to train the model contains noise.
>     - Noisy model learning: given a pre-trained model trained on noisy datasets (which we cannot access, such as the black-box pre-training data of CLIP), we want to reduce such effect brought by the noisy pre-trained data on *any* downstream datasets.
>
> That being said, our noisy model learning is naturally *complementary* to noisy label learning. We have an experiment in Sec. 4.3 to show that they can be used together.
>
> Additionally, we can informally consider that noisy training data can lead to noisy weights, as in NoisyTune [2], where the authors also found introducing noise to pre-trained weights could help with downstream tasks, reaching similar conclusions as our study.
>
> [1] Zhang et al. Understanding deep learning requires rethinking generalization. ICLR 2017.
>
> [2] Wu et al. NoisyTune: A Little Noise Can Help You Finetune Pretrained Language Models Better. ACL 2022.

---

> > ### Author Response · Authors · 2023-11-13
> > **Response to Reviewer 5WgV (2/2)**
> >
> > > 4. "In the paper, the authors claim that the proposed method enhances SVE and LSVR. However, Figure 5 (d) indicates that the proposed method does not yield superior LSVR compared to the LP and MLP models. Furthermore, it seems that as the noise ratio increases, LSVR does not drops significantly for all the settings."
> >
> > Thanks for mentioning this and we find Fig. 5 (d) might be difficult to interpret.
> > Here, we provide more explanations.
> > For Fig. 5 (d), the LSVR should be compared at the *trend* instead of the *absolute values* level. The expected LSVR across different noise ratios should be either increase or stay at the same level (it should not drop as in LP).
> > Indeed, once adding the MLP, the LSVR are already improved and stays at the same level across different noise ratio.
> > Adding the proposed regularization terms could further improve the LSVR of YFCC15M CLIP pre-trained as an increasing trend, while the ImageNet pre-trained stays at the same level.
> > This is reflected in the OOD results in Table 1 and Table 2 too, where adding MLP already improves the OOD performance, and adding regularization terms could further boost the performance.
> >
> > > 5. "It's worth noting that some related work, such as [R1], also employs Singular Value Decomposition (SVD) to address noisy label problems. It would be beneficial for the authors to discuss the distinctions between your approach and previous work."
> >
> > We would like to discuss this related work more and have included it in our related work.
> >
> > The distinctions are:
> > - We study different problems. [R1] handles the noisy label learning problem, whereas we address the noisy model learning problem, which is a rather new topic.
> > - The methods are different. [R1] utilizes the singular vectors of SVD to *filter* downstream label noise, which falls into the traditional paradigm of noise correction for noisy label learning. While we do not have such filter operation: we utilize the singular values of the SVD to encourage a better feature space and mitigate the effect of pre-training noise.
> >
> > [R1]: FINE Samples for Learning with Noisy Labels
> >
> > > 6. "I would like to point out that the improvements achieved by the proposed method appear to be relatively modest. According to the experimental results, the proposed method only demonstrates an approximately 1% improvement compared to the MLP model."
> >
> > The results we reported are the *average* performance of multiple datasets.
> > Besides, the pre-trained backbones are frozen, where we only train the linear classifier or the MLP.
> > Within this context, we do believe the improvement of approximately 1% compared to MLP tuning is significant.
> >
> > > 7. "For ResNet-50, it might be worthwhile to explore the feasibility of fine-tuning all layers rather than constraining the encoder. It would be valuable if the authors could conduct experiments to determine if the proposed loss function is effective in cases where all layers are fine-tuned."
> >
> > Thanks for your suggestion on full fine-tuning. We in general assume the pre-trained foundation models are not accessible in our study, which is mostly the case in practical large models.
> > To answer your question, we further run the full fine-tuning on R50 and showed that the proposed method is still effective.
> > Also note that in the studied datasets here, the plain fine-tuning (without our method) of R50 also present similar observation of ID performance to LP.
> > The ablation results of YFCC15M noisy pre-trained R50 are shown as follows:
> >
> > | RenderedSST2 |   0%  |   5%  |  10%  |  20%  |  30%  |
> > |:------------:|:-----:|:-----:|:-----:|:-----:|:-----:|
> > |    Full FT   | 56.62 | 58.10 | 57.81 | 57.77 | 57.57 |
> > |     Ours     | 60.39 | 60.32 | 60.04 | 59.13 | 58.96 |
> > |              |       |       |       |       |       |
> > |  **Caltech101**  |   0%  |   5%  |  10%  |  20%  |  30%  |
> > |    Full FT   | 76.97 | 77.36 | 76.25 | 76.81 | 76.24 |
> > |     Ours     | 78.24 | 78.01 | 77.92 | 77.45 | 77.19 |
> > |              |       |       |       |       |       |
> > | **FGVCAirCraft** |   0%  |   5%  |  10%  |  20%  |  30%  |
> > |    Full FT   | 33.56 | 34.24 | 34.03 | 34.39 | 33.40 |
> > |     Ours     | 36.44 | 36.53 | 36.41 | 36.27 | 36.05 |

---

> > > ### Comment · Reviewer_5WgV · 2023-11-22
> > > **Thanks for the response**
> > >
> > > I thank the authors for providing a comprehensive response with special appreciation for clarfiying the calculation of SVE and LSVR. The majority of my concerns have been successfully addressed. The incorporation of experiments encompassing fine-tuning across all layers suggests the broad applicability of the proposed loss.
> > >
> > > In summary, I believe this paper delves into a crucial and relatively novel research area. The identified phenomenon, particularly the intriguing observation that "a slight amount of noise can benefit in-domain tasks," holds substantial interest and may deserve further exploration for its potential in feature learning. Consequently, I have upgraded my score to "accept."

---

> > > > ### Author Response · Authors · 2023-11-22
> > > >
> > > > We would like to thank the reviewer for supporting our paper and glad that our response resolved your concerns. We thank you for your constructive comments to make this paper better. Indeed, the observation that "a little noise benefits ID performance" is interesting and worth further research. In the future, we will conduct more and deeper analysis towards this observation.

---

### Official Review · Reviewer_jd9i · 2023-11-01

**Soundness:** 3 good
**Presentation:** 3 good
**Contribution:** 3 good
**Rating:** 8
**Confidence:** 4

**Summary:**

The paper addresses the challenges posed by label noise in pre-training datasets and its impact on downstream tasks. The authors focus on supervised pre-training models using synthetic noisy ImageNet-1K and YFCC15M datasets. They observe that while slight noise in pre-training can enhance in-domain (ID) transfer performance, it consistently harms out-of-domain (OOD) performance. The reason behind is noise in pre-training shapes the feature space differently. They introduce a lightweight black-box tuning method, NMTune, to mitigate the adverse effects of noise and improve generalization on both ID and OOD tasks.

**Strengths:**

- This paper studies a problem that is both practical and significant, yet has not been sufficiently investigated in prior research.
- This paper is well-motivated and easy to follow.
- The analysis of features is useful to understand the noise's impact on ID and OOD data.
- Experiments are comprehensive.

**Weaknesses:**

- Improvements are needed in the writing and presentation quality. Please check the Question section below.
- Some claims in the paper are unclear and confusing. Please check the Question section below.
- The authors did not mention the limitations of their method and potential future work. The paper does not explore or discuss potential failure cases of the proposed methods. Understanding when and why the methods might fail is crucial for practical applications

**Questions:**

- Self-supervised pre-train does require external supervision. Does it mean those models will not suffer from the noise issue?
- Does it proposed method generalize to self-supervised pre-trained models?
- When you trained CLIP, did you train the text encoder together or you use a frozen text encoder?
- "For OOD evaluation, we use DomainNet (Peng et al., 2019) where we train on either “real” or “sketch” images and test on “real”, “sketch”, “inpainting”, and “clippart” images". If you trained on either “real” or “sketch”, you should only test on domains that the model did not seen during training right？ This sentence is a bit confusing.
- "we empirically analyze the singular value spectrum of the pre-trained **the** feature space on downstream datasets" Typo: extra "the"
- Section 2.3,  the authors should let or remind the readers what are M and D. They should be the number of samples and latent dimension size right?
- Figure 3 is a bit confusing. For a specific noise level, there are many points (e.g. many blue stars). What does each point mean? One downstream task? A lot of points are overlapped and I don’t know which 5 points (0%, 5%, 10%, 20%, 30%) should be read together.
- "An initial increase in the spanning dimension of the feature space is beneficial to the discriminability on ID tasks. " The reason behind that is "the pre-trained feature extractor captures more structure in data" due to noise. But why more structure does not help OOD?

---

> ### Author Response · Authors · 2023-11-13
> **Response to Reviewer jd9i (1/2)**
>
> > 1. "Improvements are needed in the writing and presentation quality.", "Some claims in the paper are unclear and confusing."
>
> We sincerely thank you for your careful proofreading of our paper.
> We have fixed the typos and made the confusing statement more clear in the revised paper.
>
> > 2. "The authors did not mention the limitations of...why the methods might fail is crucial for practical applications"
>
> Thanks for pointing this out. We briefly mentioned the limitation in the conclusion section (the last two sentences) due to the space limit.
> Now we discuss more limitation and potential failure here, which has been added in appendix C.1 and C.2. of the paper.
>
> **Limitation**. The limitation mainly lies in our empirical study of the noise in pre-training.
> Due to the limited computing resources to extensively investigate different noise ratios by pre-training different models, we could only conduct our "motivational experiments" on ImageNet-1K and YFCC15M using ResNet-50 (which are already very expensive since we need to pre-train several ResNet-50 models on ImageNet or YFCC15M from scratch).
> Note that most of the SOTA foundation models are of much more parameters and are trained on much larger (and possibly inaccessible) datasets, which are beyond our reach to perform pre-training.
> Additionally, the empirical experiments are limited to actual supervised pre-training.
> While there are some existing studies showing the potential issue of noisy pre-training data, we could reveal more by scaling our experiments on larger backbones and datasets in the future.
>
> **Potential Failure**. Yes, we agree that identifying failure cases indeed helps understand the problem better. We do observe some failure cases of the proposed methods.
> For example, from the results in Table 7, the proposed method falls short of LP on Caltech101 on almost all backbones we studied, while improving over MLP.
> Our hypothesis for the failure is that the SVD regularization term in the proposed method might need to optimize the top-$K$ singular values instead of just the largest one. The optimal value of $K$ might be different across datasets. However, setting $K=1$ can already achieve reasonable performance for most of the tasks, as shown in the results.
>
>
> > 3. "Self-supervised pre-train does require external supervision...to self-supervised pre-trained models?"
>
> Thanks for this wonderful question! We do believe self-supervised learning is worth further discussion:
>
> - First of all, self-supervised learning does not require the explicit supervision of $Y$, but can be seen as supervised by the original input $X$, as we mentioned in footnote 2 of page 2 in our paper. In this context, it still requires supervision and the noise problem still exists: if $X$ is corrupted, which is usually the case in large-scale pre-training, it can be viewed as noise in self-supervision.
> - Second, despite some previous research [1] indicating that the self-supervised pre-training data contains some noise, we give more examples here. It turns out that corruption in $X$ can also have different formats in the actual data. For instance, in images, the corrupted $X$ can be poisoned or attacked images that do not contain natural content, and it will become noise for masked image modeling pre-training. Taking contrastive learning, MOCO [2], as another example. MOCO is a instance discrimination task and the negative pairs could potentially contain images that are similar to the positive anchor. This can also be viewed as noise. In auto-regressive language models such as GPT series, the noise would simply wrong or repetitive words in the sentence.
> - Third, our algorithm can naturally work for self-supervised pre-training models, as shown in Table 1 and 2, where most of these foundation models (ViT, CLIP, EfficientNet, GPT2, BERT, RoBERTa, and text-ada-002) are self-supervised pre-trained. Studying and understanding the corrupted $X$ (or noise) in self-supervised pre-training would also be very interesting and is our future direction. We believe the observation and conclusion from this work would scale and extend to the self-supervised learning scenarios because eventually, they fall into the same general formulation of noise in supervision.
>
> In the future version of the paper, we will add more experiments and discussions about self-supervised learning.
>
> [1] Yang et al. Xlnet: Generalized autoregressive pretraining for language understanding. NeurIPS 2019.
> [2] Kaiming He, et al. Momentum Contrast for Unsupervised Visual Representation Learning
>
> > 4. "When you trained CLIP, did you train the text encoder together or you use a frozen text encoder?"
>
> For CLIP pre-training, we strictly follow the training recipe of the Resnet-50 model in open clip, thus both the image encoder and text encoder are trained from scratch on synthetic noisy YFCC15M.
> For CLIP fine-tuning (i.e., our noisy model tuning), we fixed all the encoders.

---

> > ### Author Response · Authors · 2023-11-13
> > **Response to Reviewer jd9i (2/2)**
> >
> > > 5. "For OOD evaluation, we use DomainNet (Peng et al., 2019) where we train on either “real” or “sketch” images and test on “real”, “sketch”, “inpainting”, and “clippart” images". If you trained on either “real” or “sketch”, you should only test on domains that the model did not seen during training right？ This sentence is a bit confusing.
> >
> > Sorry for this confusing statement and we definitely do not break the rule in OOD testing. We train either on "real" or "sketch" split and test on the unseen domains:
> > - If we train on "real", we test OOD performance on “sketch”, “inpainting”, and "clippart".
> > - If we train on "sketch", we test OOD performance on "real", “inpainting”, and "clippart".
> >
> > This is similar for the OOD evaluation of ImageNet variants, where we train on ImageNet, and test on other variants such as ImageNet-A and ImageNet-R.
> > We have made this more clear in the revised paper.
> >
> > > 6. "we empirically analyze the singular value spectrum of the pre-trained the feature space on downstream datasets" Typo: extra "the"
> >
> > Thanks for pointing this out. We have fixed this typo in the revision.
> >
> >
> > > 7. "Section 2.3, the authors should let or remind the readers what are M and D. They should be the number of samples and latent dimension size right?"
> >
> >
> > Thanks for this kind reminder. Here $M$ means the downstream dataset size (number of samples), and D denotes the feature dimensions.
> > We have added a footnote about this in the revised paper.
> >
> > > 8. "Figure 3 is a bit confusing. For a specific noise level, there are many points (e.g. many blue stars). What does each point mean? One downstream task? A lot of points are overlapped and I don’t know which 5 points (0%, 5%, 10%, 20%, 30%) should be read together."
> >
> > We are sorry for the confusion in Fig. 3 and here we provide more explanations.
> >
> > First, each color and marker represents a different pre-training noise ratio. We plot the average accuracy of different percentages of downstream datasets and the SVD (or LSVR) of the downstream test data for each downstream task. Thus each point corresponds to a downstream task. The results of different pre-training noise ratios for each task are thus clustered together.
> >
> > We have made an update to Fig. 3(a) and Fig. 3(c) to provide a zoom-in view of the figure to present the trend we found, as shown in Fig. 10 of Appendix.
> > We have also included the above explanation in Appendix A.4.
> >
> >
> > > 9. "An initial increase in the spanning dimension of the feature space is beneficial to the discriminability on ID tasks. " The reason behind that is "the pre-trained feature extractor captures more structure in data" due to noise. But why more structure does not help OOD?"
> >
> >
> > The SVE (more structure in data) and LSVR should be looked at together to interpret this.
> > As the noise ratio increases in the pre-training data, the model learns to fit more noise in pre-training.
> > This will result in the pre-trained feature space with larger SVE, thus capturing more structure on downstream data.
> > Initially increase in SVE can help with ID downstreams because the nuance components related to the smallest singular values are not affected too much by slight noise.
> > But as noise increases, the singular values of the nuance components also get increase, resulting in worse ID performance.
> > Also, larger SVE comes at the cost of smaller LSVR, as the model uses more capacity/structure/dimension in feature space to fit noise, the generalizable components related to the top singular values are getting decreased, with smaller LSVR presented and worse OOD performance.
> > The change in SVD spectrum is shown in the Fig.11 - Fig.14 in Appendix.

---

> > > ### Comment · Reviewer_jd9i · 2023-11-22
> > > **Thanks for the responds**
> > >
> > > The authors have addressed most of my questions and improved the manuscript. I will increase my score to 8.

---

> > > > ### Author Response · Authors · 2023-11-22
> > > >
> > > > We appreciate the valuable suggestions and feedback from the reviewer. We are also glad that most of your concerns have been addressed. Thanks again for increasing the rating!

---

### Author Response · Authors · 2023-11-13
**General Response**

We first thank for all reviewers for pointing out the strength of our work:
- the problem we studied is novel, practical, and significant
- with extensive experiment and intriguing analysis
- the proposed method is intuitive and effective

As for the confusion of Fig. 3, we want to make some clarifications:
- We have added the details of how we plot Fig. 3 in the Appendix.
- We have added a better visualization in Fig. 10, with the overlapped and clustered points being zoomed in.

Finally, we submitted the code for reproducibility.

We hope the following individual response could resolve the concerns and answer the questions raised by the reviewers.

---

### Meta-Review · Area_Chair_B59T · 2023-12-06

**Metareview:**

This paper studies the impact of noise in pretraining data on downstream tasks. It concludes that noise changes the feature space, resulting in, (surprisingly) sometimes beneficial, and perhaps less surprisingly, sometimes harmful effects. The authors propose a mitigation technique based on a set of regularizations.

This is a largely empirical paper that has pretty extensive evaluations offers interesting insights. The reviewers all agree, and have additionally clarified minor points of confusion. There are few weakness/missing aspects. One question, which came up, is whether these results really hold up for much larger models, but this is not a reasonable ask due to the cost of running these types of experiments.

**Justification For Why Not Higher Score:**

While this is a strong paper, it does not have the level of novelty/originality or potential impact that I would expect to see for an oral.

**Justification For Why Not Lower Score:**

Generally, both the reviewers and I enjoyed the work and found it potential useful, perhaps beyond the level of a bare accept.

---

### Decision · Program_Chairs · 2024-01-16

Accept (spotlight)